# Piezo1/2 mediate mechanotransduction essential for bone formation through concerted activation of NFAT-YAP1-ß-catenin

Taifeng Zhou[1,2], Bo Gao[1,3], Yi Fan[1], Yuchen Liu[1], Shuhao Feng[1,4], Qian Cong[1], Xiaolei Zhang[1,5], Yaxing Zhou[1], Prem S Yadav[1], Jiachen Lin[1,6], Nan Wu[6], Liang Zhao[4], Dongsheng Huang[3], Shuanhu Zhou[7], Peiqiang Su[2]*, Yingzi Yang[1]*

[1]Department of Developmental Biology, Harvard School of Dental Medicine, Harvard Stem Cell Institute, Boston, United States; [2]Department of Orthopaedic Surgery, Guangdong Provincial Key Laboratory of Orthopedics and Traumatology, First Affiliated Hospital of Sun Yat-sen University, Sun Yat-sen University, Guangzhou, China; [3]Department of Spine Surgery, Sun Yat-sen Memorial Hospital, Sun Yat-sen University, Guangzhou, China; [4]Department of Orthopedic Surgery, Nanfang Hospital, Southern Medical University, Guangdong, China; [5]Department of Operative Dentistry and Endodontics, Guanghua School of Stomatology, Sun Yat-sen University, Guangzhou, China; [6]Department of Orthopedic Surgery and Beijing Key Laboratory for Genetic Research of Skeletal Deformity, Peking Union Medical College Hospital, Chinese Academy of Medical Sciences, Beijing, China; [7]Department of Orthopedic Surgery, Brigham and Women's Hospital, Boston, United States

*For correspondence:
supq@mail.sysu.edu.cn (PS);
yingzi_yang@hsdm.harvard.edu
(YY)

Competing interests: The authors declare that no competing interests exist.

**Abstract** Mechanical forces are fundamental regulators of cell behaviors. However, molecular regulation of mechanotransduction remain poorly understood. Here, we identified the mechanosensitive channels Piezo1 and Piezo2 as key force sensors required for bone development and osteoblast differentiation. Loss of Piezo1, or more severely Piezo1/2, in mesenchymal or osteoblast progenitor cells, led to multiple spontaneous bone fractures in newborn mice due to inhibition of osteoblast differentiation and increased bone resorption. In addition, loss of Piezo1/2 rendered resistant to further bone loss caused by unloading in both bone development and homeostasis. Mechanistically, Piezo1/2 relayed fluid shear stress and extracellular matrix stiffness signals to activate $Ca^{2+}$ influx to stimulate Calcineurin, which promotes concerted activation of NFATc1, YAP1 and ß-catenin transcription factors by inducing their dephosphorylation as well as NFAT/YAP1/ß-catenin complex formation. Yap1 and ß-catenin activities were reduced in the Piezo1 and Piezo1/2 mutant bones and such defects were partially rescued by enhanced ß-catenin activities.

## Introduction

Mechanical forces are part of environmental cues that are sensed and responded to during embryonic development and adult life for proper morphogenesis and tissue/organ functions. Apart from the well-known complex networks of biochemical signaling that direct the development of higher vertebrate embryos, cells also constantly receive mechanical information from their environment in the form of biophysical stimuli such as stress, strain and fluid flow, which are generated by gravity,

cell movement and cell-cell or cell-extracellular matrix (ECM) interactions. The process of mechano-transduction, conversion of mechanical forces into biological signals, is a fundamental physiologic process critical for revealing environmental features to almost all cells in an organism. However, despite the long-recognized key regulatory roles of mechanotransduction for embryonic development and sensory perception (*Wozniak and Chen, 2009*), little concrete information on the molecular mechanisms that integrate biophysical stimuli with gene regulation to guide fundamental events such as cell fate determination, proliferation and apoptosis is known. Among all organs, mechano-transduction is especially important in the skeleton system as its development and physiological functions require consistently sensing and responding to mechanical forces. A range of human conditions in which fetal movement is diminished were found to impact severely on skeletal development (*Nowlan et al., 2010a*). For instance, diseases of the neuromuscular system, such as congenital myotonic dystrophy (*Wesström et al., 1986*) or spinal muscle atrophy (*Nicole et al., 2002*) have a dramatic effect on skeletal development leading to smaller, thinner and weaker long bones, prone to postnatal fracture (*Rodríguez et al., 1988a*; *Rodríguez et al., 1988b*). Indeed, embryonic bone formation was reduced in experimentally induced muscle paralysis in chick (*Hall and Herring, 1990*) or genetically engineered 'muscleless/reduced muscle' mouse mutant embryos (*Nowlan et al., 2010b*). In postnatal life, rapid loss of bone mass and even osteoporosis in some cases due to a lack of load caused by prolonged bed rest, spinal cord injury or space flight and the bone anabolic effects of exercise are well documented. The major skeletal cells such as mesenchymal stem cells and the derived osteoblasts, chondrocytes and osteocytes are mechanical receptor cells that perceive and respond to forces, such as those generated by cell-ECM interactions or fluid shear stress (FSS). Ion channels including the transient receptor potential (TRP) polycystin family members have been found to regulate bone development (*Katsianou et al., 2018*). However, the molecular mechanisms whereby mechanical signals are sensed and converted by skeletal cells to biochemical signals leading to transcription factor activation (*Huang and Ogawa, 2010*) and eventually specific cellular responses such as cell fate determination (*Duncan and Turner, 1995*) remain largely unknown.

Recent advancement in mechanotransduction research (*Ingber, 2006*; *Kelly and Jacobs, 2010*; *McMahon et al., 2008*; *Halder et al., 2012*; *Humphrey et al., 2014*) has opened the door to providing molecular understanding of mechanical force sensing and downstream regulatory events in multiple biological systems. The mechanosensitive (MS) Piezo1 and Piezo2 channels are to date the best characterized biological force-sensing systems (*Parpaite and Coste, 2017*) that are involved in processes as diverse as perceiving touch or regulating the volume of red blood cells in mammals (*Martinac, 2012*; *Coste et al., 2012*; *Murthy et al., 2017*). The transcription factors Yap1 and Wwtr1 (also known as TAZ) in the Hippo signaling pathway have been identified as essential effectors of mechanotransduction (*Halder et al., 2012*; *Aragona et al., 2013*; *Codelia et al., 2014*; *Wada et al., 2011*; *Dupont et al., 2011*). The Piezo channels are the first type of MS channels documented to underlie a human disease linked to mechanical pathologies including a number of blood disorders and problems with proprioception (*Chesler et al., 2016*). While recent studies showed that Piezo1 in osteoblast cells and osteocytes are required for bone formation and regulating bone resorption in postnatal mice (*Sun et al., 2019*; *Li et al., 2019*; *Wang et al., 2020*), we additionally investigated in this study the functions and molecular mechanisms of both Piezo 1 and 2 channels in embryonic skeletal development and bone marrow stromal cells (BMSCs). Our work identifies Piezo1/2 and the downstream mechanotransduction pathways as key regulators that link mechanical microenvironment to a signal transduction cascade leading to nuclear transcriptional changes to promote bone formation in development and osteoblast differentiation from stem cells. Our studies provide mechanistic insights to decipher the functions of PIEZO1 in human skeleton. Human PIEZO1 SNPs are associated with body height reduction (*Marouli et al., 2017*). In a recent estimated bone mineral density (eBMD) GWAS of 426,824 individuals in the UK Biobank (*Morris et al., 2019*), novel genetic influences on osteoporosis in humans were identified with two SNPs (rs4238686, rs11643303) in PIEZO1 among the 1103 conditionally independent signals (423 novel) at genome-wide significance (*Supplementary file 2* Table 2 *Morris et al., 2019*).

## Results

### Piezo 1 and 2 are expressed in the developing long bones

To test our hypothesis that the MS Piezo channels are key mechanosensors required for skeletal development and homeostasis, we first examined their expression in the developing limb buds, in which long bones form through endochondral ossification. At E12.5 to E14.5, when endochondral bone initiates, we found by whole mount in situ hybridization (WISH) that *Piezo1* is mostly expressed in the interdigit region while *Piezo2* is expressed in the forming digit and wrist (*Figure 1a*, *Figure 1—figure supplement 1a,b*). To further determine the expression of *Piezo1* in the developing long bones, we performed in situ hybridization with *Piezo1* probes using the RNAscope technology on sections (*Wang et al., 2012*). *Piezo1* was most strongly expressed in the connective tissues associated with the muscle and weaker expression of *Piezo1* expression was detected in the muscle and differentiating osteoblast cells in the perichondrium and periosteum (*Figure 1—figure supplement 1c*). The weak *Piezo1* expression in the skeletal tissue prompted us to determine Piezo1 protein expression utilizing the *Piezo1*$^{P1-tdT}$ mice that allow sensitive detection of Piezo1 protein in vivo by expressing a C-terminus fusion protein of Piezo1 with the fluorescent tdTomato reporter from the *Piezo1* locus (*Ranade et al., 2014*). As the direct red fluorescent signal was weak, we used anti-RFP antibodies to detect tdTomato (*Figure 1b,c*). Consistent with the in situ hybridization data (*Figure 1—figure supplement 1c*), Piezo1 protein was detected in the connective tissue, the associated muscle and differentiating osteoblast cells that express Osterix (Sp7) in the perichondrium and periosteum at E13.5, E15.5 and postnatal day 0 (P0) neonatal pups (*Figure 1b,c*). Piezo1 expression was also detected in the hypertrophic chondrocytes, tendons and ligaments (*Figure 1b,c*). To detect Piezo2 protein expression, we took an indirect approach using the *Piezo2-EGFP-IRES-Cre* (*Piezo2*$^{EGFP-Cre}$) mice that express Piezo2-EGFP-Cre fusion protein from the endogenous *Piezo2* locus (*Woo et al., 2014*). The EGFP signal was too weak to be reliably detected, so we crossed the *Piezo2*$^{EGFP-Cre}$ mouse with a Rosa-tdTomato reporter mouse (JAX stock # 007914). The tdTomato$^+$ Piezo2 lineage cells (Piezo2$^+$ cells and their descendants) included Sp7$^+$ osteoblast cells, joint cells, sporadic growth plate chondrocytes, tendons and connective tissue cells in the muscle (*Figure 1d*, *Figure 1—figure supplement 1d,e*). Consistent with previous studies, Piezo2-derived tdTomato$^+$ cells were found in the skin (*Figure 1d*). Taken together, these results show that both Piezo1 and 2 were expressed mostly in the differentiating osteoblasts in the developing skeleton and Piezo1 was also expressed in hypertrophic chondrocytes, suggesting that they may regulate mechanotransduction during bone development.

### Piezo 1 and 2 are required for bone formation and long bone growth

To investigate whether *Piezo1/2* regulate skeletal development in vivo, we removed *Piezo1* and/or *Piezo2* in early limb bud mesenchyme before skeletogenesis starts using the floxed *Piezo1* and *Piezo2* lines (*Woo et al., 2014*; *Cahalan et al., 2015*) and the *Prrx1-Cre* line (*Logan et al., 2002*; *Figure 2*, *Figure 2—source data 1*). While both *Piezo1*$^{-/-}$ and *Piezo2*$^{-/-}$ mutants exhibit embryonic lethality, the *Prrx1-Cre; Piezo1*$^{f/f}$ (*Piezo1* CKO) and *Prrx1-Cre; Piezo2*$^{f/f}$ (*Piezo2* CKO) mice were viable at birth. However, the *Piezo1* CKO mice exhibited multiple bone fractures in radius and ulna suggesting severely reduced bone formation (*Figure 2a*). The *Piezo2* CKO mice exhibited grossly normal skeletal development with no bone fracture, but the *Piezo1/2* double conditional (*Piezo1/2* DKO) mutant mice demonstrated more severe skeletal defects including additional fractures in the femur (*Figure 2a,b*, *Figure 2—source data 1*, *Figure 2—figure supplement 1a*, *Figure 2—figure supplement 1—source data 1*). We also found that long bones were shortened in the absence of *Piezo1*, but not *Piezo 2*, and further shortening was observed in the *Piezo1/2* DKO mutants (*Figure 2b*). These results indicate that while *Piezo1* plays a major role, *Piezo2* shares some of the *Piezo1* functions in mesenchymal progenitor cells during skeletal development.

Mechanical stress increases postnatally and we found that Piezo1 expression was progressively increased in young mice (*Figure 2—figure supplement 1b*), suggesting that Piezo1 also regulates bone formation after birth in adult lives. However, *Piezo2* mRNA expression was reduced in postnatal mice in the cortical bone (*Figure 2—figure supplement 1b*). The μCT analyses showed that both cortical and trabecular bone masses were reduced in the *Piezo1* CKO mice and further reduced in the *Piezo1/2* DKO mice at the age of P21. The *Piezo2* CKO mice exhibited normal bone mass

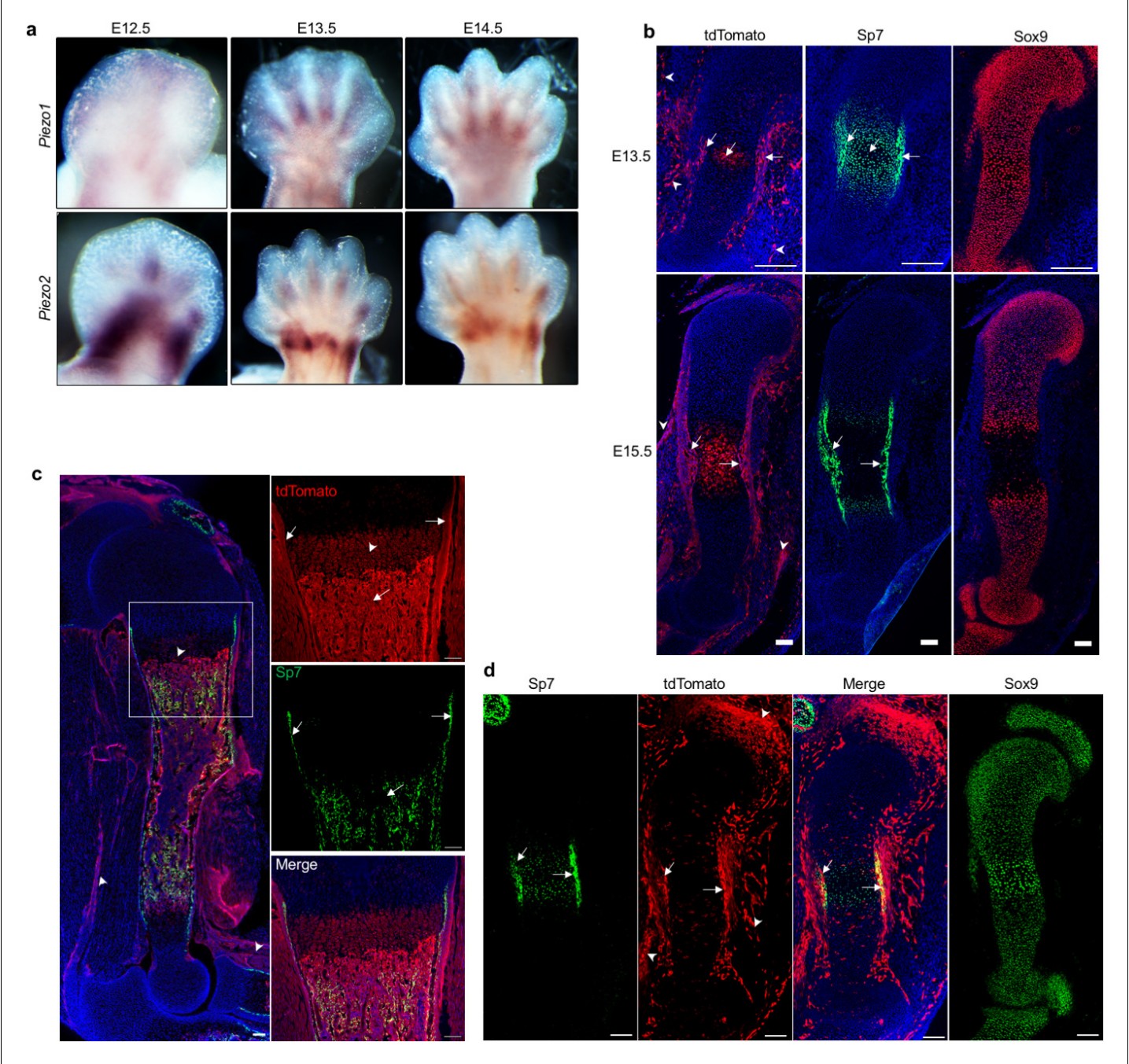

**Figure 1.** Characterization of Piezo1 and Piezo2 expression in the developing long bones. (**a**) Expression of *Piezo1* and *Piezo2* examined by whole-mount in situ hybridization of embryonic limb buds at the indicated stages. (**b**) Immunostaining for tdTomato, Sp7 and Sox9 in consecutive humerus sections of the Piezo1-tdTomato forelimb buds. Arrows: Membranous tdTomato and nuclear Sp7 were found in the differentiating osteoblasts and hypertrophic chondrocytes in the same regions. Arrowheads: Extra-skeletal Piezo1-tdTomato expression. (**c**) Representative image of GFP and tdTomato fluorescent staining of humerus sections from the Piezo1-tdTomato; Sp7-Cre::GFP mouse at P0. The boxed region was shown in higher magnification in the right panel. Arrows: colocalization of tdTomato and GFP in the differentiating osteoblasts. Arrowheads: Hypertrophic chondrocyte or extra-skeletal expression of Piezo1-tdTomato. (**d**) Representative images of tdTomato costained with Sp7 or Sox9 on consecutive humerus sections from the E13.5 *Piezo2Cre; Rosa26-tdTomato* limb bud. Arrows: TdTomato costained with Sp7 in the differentiating osteoblasts. Arrowheads: Joint or extra-skeletal Piezo1-tdTomato expression. All scale bars, 100 μm. DAPI (blue) stain the nucleus.

The online version of this article includes the following figure supplement(s) for figure 1:

**Figure supplement 1.** Mouse *Piezo1* and *Piezo2* expression.

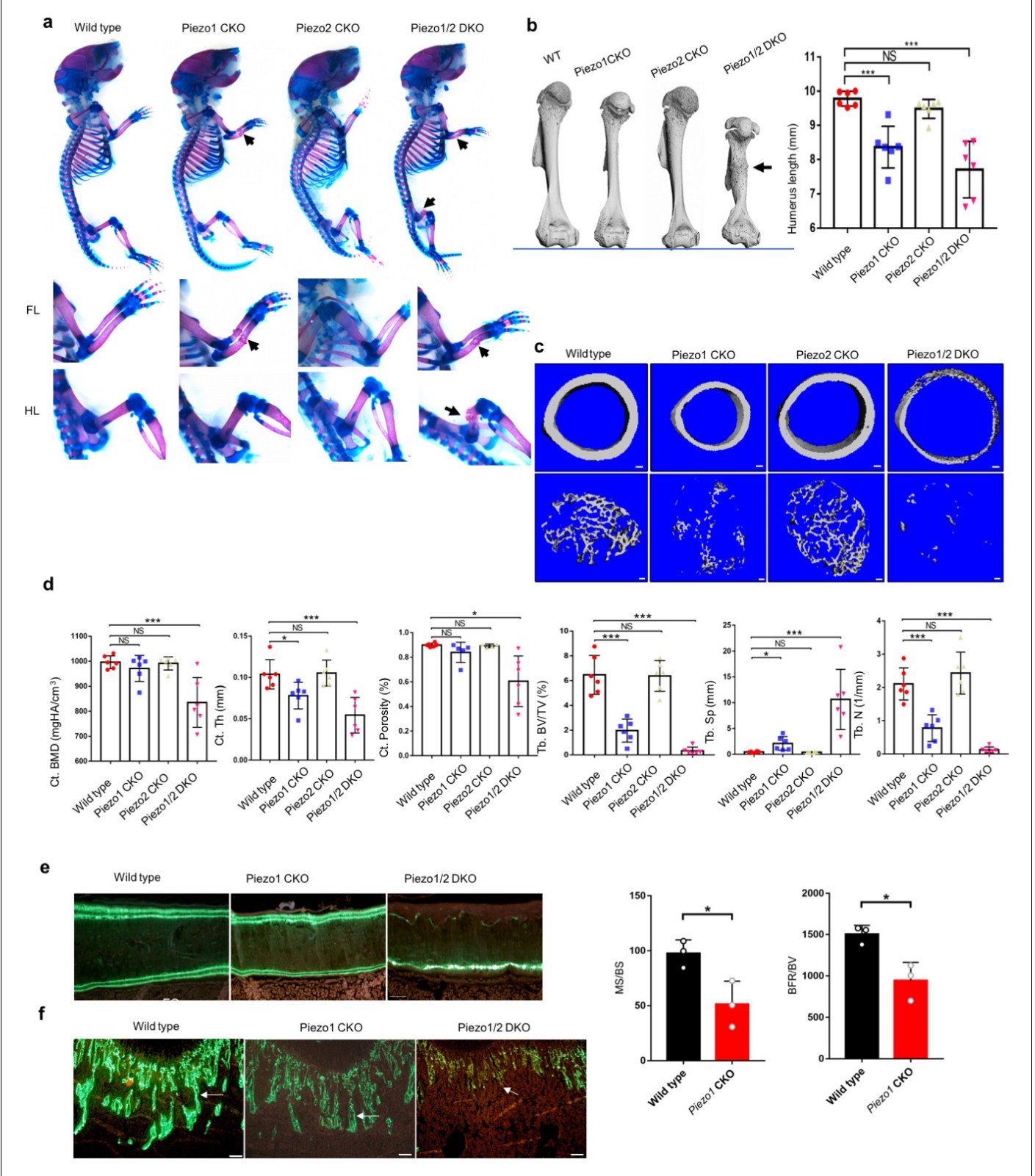

**Figure 2.** Loss of *Piezo1* or both *Piezo1/2* in embryonic limb mesenchyme led to reduced bone formation and spontaneous bone fractures in neonatal mice. (a) Whole mount alizarin red and alcian blue staining of P0 mouse pups from the same litter. Bone fractures were indicated by arrows. The forelimb (FL) and hindlimb (HL) were taken out and shown in the lower panel. (b) Representative three dimensional μCT images of humerus from 3 weeks old littermate mice with the indicated genotypes. The humerus length was quantified (n = 6, mean ± SD). Bone fracture is indicated by an arrow.

*Figure 2 continued on next page*

*Figure 2 continued*

(c) Representative cross section µCT images of the cortical and trabecular femur bones from 3 weeks old littermate mice. (d) Quantified analysis of µCT data. Data are shown as means ± SD. (e, f) Histomorphometric analysis of distal femurs from wild type and *Prrx-Cre* driven *Piezo1* CKO and *Piezo1/2* DKO mutant mice. Representative images of double Calcein labeling in the cortical (e) and trabecular (f) bones were showed, and the dynamic bone formation parameters were only quantifiable in the wild type and *Piezo1* CKO group, due to the severely reduced cortical bone formation in the *Piezo1/2* DKO and little bone formation in the secondary spongiosa of the distal metaphysis in both *Piezo1* CKO and *Piezo1/2* DKO mutant mice. PO: periosteum; EO; endosteum. Scale bars: 50 µm (e); 100 µm (f). *p<0.05, **p<0.01, ***p<0.001, one-way ANOVA followed by Tukey's multiple comparisons tests. In (e), p value is calculated by two-tailed unpaired Student's *t*-test (***Source data 1***).

The online version of this article includes the following source data and figure supplement(s) for figure 2:

**Source data 1.** Loss of *Piezo1* or both *Piezo1/2* in embryonic limb mesenchyme led to reduced cartilage growth and osteoblast differentiation.
**Figure supplement 1.** Gross characterization of *Prrx1-Cre* driven *Piezo1 and Piezo2* mutant phenotypes at postnatal stages.
**Figure supplement 1—source data 1.** Original numbers used for quantification and Western blots.

(***Figure 2c,d***, ***Figure 2—source data 1***). The *Prrx1-Cre; Piezo1^{f/+}; Piezo2^{f/+}* embryos or mice also did not show phenotypic difference compared with wild-type controls. We therefore focused our analyses on the *Piezo1* CKO and *Piezo1/2* DKO embryos and newborn mice. Analyses of postnatal bone development of *Piezo1/2* mutant pups were performed by Masson's Trichrome staining, which showed reduction of collagens in the trabecular and cortical bones of the P5 *Piezo1* CKO and *Piezo1/2* DKO mutant pups (***Figure 2—figure supplement 1c***). Bone formation was definitively examined by ELISA assays of serum procollagen type I N propeptide (PINP) levels (***Figure 2—figure supplement 1c***), which is considered clinically as the most sensitive bone formation marker (***Samoszuk et al., 2008***). Furthermore, histomorphometric analysis showed reduced mineralization apposition rate (MAR), and bone formation rate (BFR) in periosteum of the cortical bone of the *Piezo1* CKO mice (***Figure 2e***, ***Figure 2—source data 1***). Reduction of cortical bone formation in the *Piezo1/2* DKO mutant mice or trabecular bone formation in both *Piezo1* CKO and *Piezo1/2* DKO mice were too severe to allow accurate quantification (***Figure 2e,f***). In the *Piezo1/2* DKO mice, the secondary spongiosa was almost completely missing.

Reduction of bone formation in the *Piezo1* CKO and *Piezo1/2* DKO mutant mice was accompanied by increase in bone remodeling indicated by upregulated expression of osteoclast markers Cathepsin K (Ctsk) and Tartrate-resistant acid phosphatase (Acp5 or Trap) (***Figure 2—figure supplement 1e,f***). However, earlier in development at E16.5, there was no difference in osteoclast marker expression (data not shown) suggesting that increased osteoclast differentiation may have resulted from the changes in osteoblast cells. Indeed, quantitative gene expression analysis by qRT-PCR further showed that while early and mature osteoblast marker expression were reduced, *Tnfsf11* (*Rankl*) and *Acp5* expression, indicative of osteoclast differentiation, were increased in the *Piezo1* CKO and *Piezo1/2* DKO mutant at P0 (***Figure 2—figure supplement 1g***). Expression of *Tnfrsf11b* (*Opg*), which encodes a decoy receptor of Tnfsf11 to inhibit osteoclast differentiation, was also reduced. Therefore, reduced bone mass and density in the absence of Piezo channels is a combination of reduced bone formation and increased bone resorption, which resulted in not only bone fractures, but also reduced longitudinal bone growth and bone deformity such as bowing of the tibia (***Figure 2—figure supplement 1a***).

To understand the cellular mechanisms underlying the skeletal defects in the *Piezo1/2 CKO* mutant mice, we first examined chondrocyte and osteoblast differentiation. At E16.5 and P0, expression of the chondrocyte marker Sox9 and hypertrophic chondrocyte marker Col10a1 was similar in the wild-type control, *Piezo1* CKO and *Piezo1/2* DKO embryos (***Figure 3a***, ***Figure 3—figure supplement 1***), suggesting cartilage formation and chondrocyte hypertrophy was not regulated by *Piezo1/2* in development. However, expression of the early osteoblast marker Sp7 was markedly reduced at E16.5 and P0 in the ossifying regions of the developing *Piezo1* CKO humerus and further reduced in the *Piezo1/2* DKO humerus (***Figure 3a*** and ***Figure 3—figure supplement 1***), suggesting that *Piezo1/2* are required to promote osteoblast differentiation in development. Bone mineralization and osteoblast maturation as indicated by von Kossa staining and secreted phosphoprotein 1 (Spp1, also known as Opn) immunofluorescent staining were also reduced in the trabecular and cortical bone areas at P0 (***Figure 3d,e***). When examined by BrdU labeling, we found that growth plate chondrocyte proliferation and the length of the proliferative regions were both reduced in the *Piezo1* CKO and further reduced in the *Piezo1/2* DKO mutant at E16.5 (***Figure 3b***, ***Figure 3—source data***

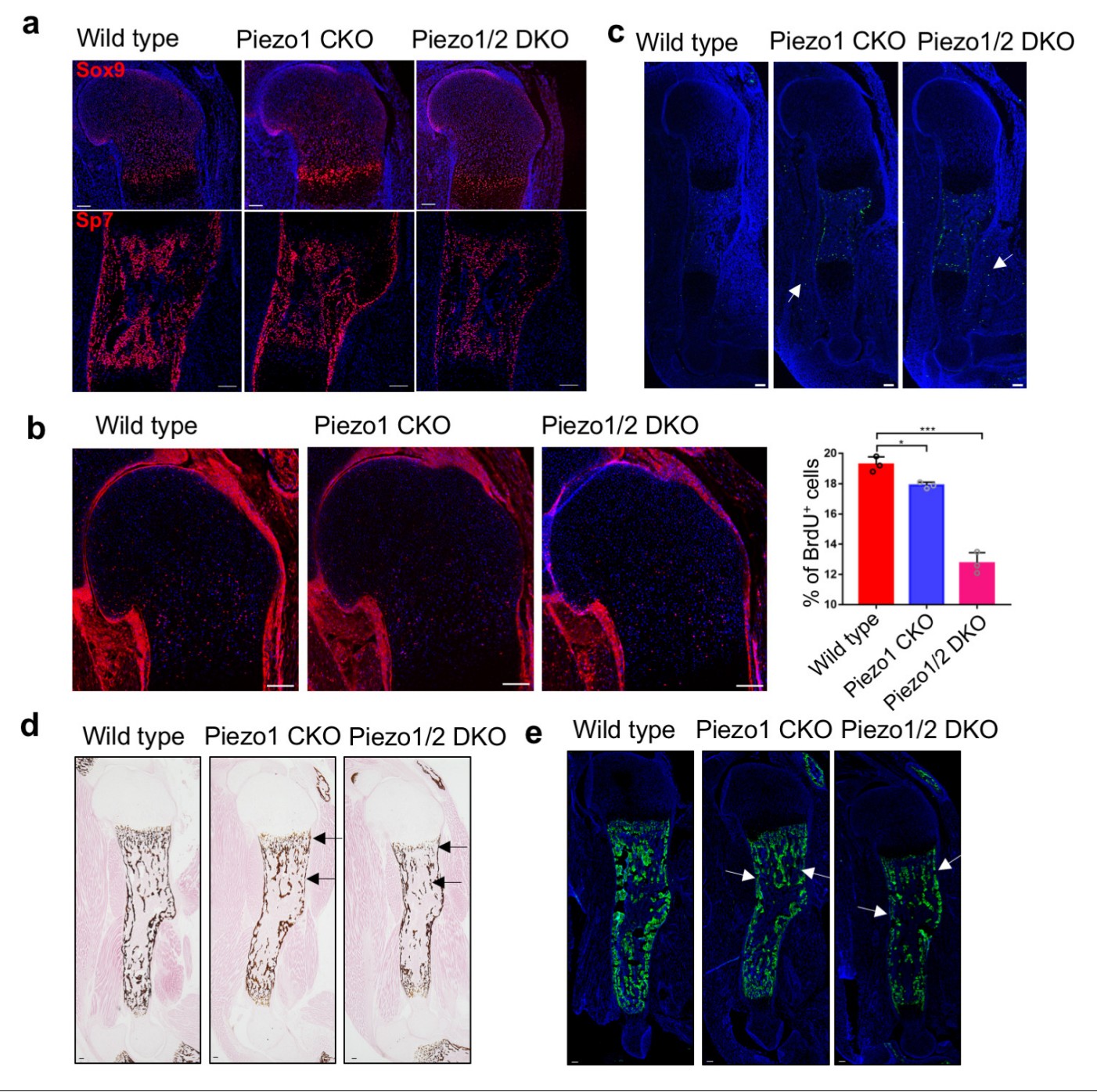

**Figure 3.** Loss of *Piezo1* or both *Piezo1/2* in embryonic limb mesenchyme led to reduced cartilage growth and osteoblast differentiation. (a) Immunofluorescent images of Sox9 and Sp7 expression in the humerus sections of E16.5 embryos. (b) Representative BrdU of humerus sections of E16.5 embryos. The percentage of BrdU$^+$ cells in the growth plate was quantified (n = 3, means ± SD). *p<0.05, **p<0.01, ***p<0.001, one-way ANOVA followed by Tukey's multiple comparisons tests. Proliferative cartilage regions were indicated by double headed arrows (*Figure 3—source data 1*). (c) Representative TUNEL staining of humerus sections of E16.5 embryos. TUNEL signals in the muscle were indicated by arrows. (d) von Kossa staining and (e) Spp1 immunostaining of humerus sections from P0 littermate pups. Reduced staining was indicated by arrows. All scale bars, 100 μm. DAPI (blue) stain the nucleus.

The online version of this article includes the following source data and figure supplement(s) for figure 3:

**Source data 1.** Quantification of cell proliferation.
**Figure supplement 1.** Immunostaining of Sox9, Col10 and Sp7 in sections of humerus cartilage from P0 pups.

1). Increased apoptosis was found in the bone and muscle of the *Piezo1* CKO and *Piezo1/2* DKO mutant at E16.5 by TUNEL assay (*Figure 3c*). These analyses indicate that Piezo1/2 MS channels are required for cartilage growth and bone formation in embryonic development, providing mechanistic insights into findings made in human populations that *PIEZO1* SNPs are associated with reduced body height and eBMD (*Marouli et al., 2017*; *Morris et al., 2019*).

## Piezo 1 and 2 are required for bone formation and maintenance by mechanical forces

To understand more specifically the role of *Piezo1/2* in bone development, we removed *Piezo1/2* using the *Sp7-Cre* mouse line (*Rodda and McMahon, 2006*; *Figure 4a–e*, *Figure 4—source data 1*). Interestingly, the *Sp7-Cre; Piezo1^{f/f}* (*Piezo1*CKO) and *Sp7-Cre; Piezo1^{f/f}; Piezo2^{f/f}* (*Piezo1/2* DKO) mouse pups did not show limb bone fractures. However, multiple bone fractures were found in the ribs at P21 in the *Piezo1CKO* and *Piezo1/2* DKO mice, but not the *Piezo2* CKO mice (*Figure 4a*). μCT analyses showed that loss of *Piezo1* or *Piezo1/2* in Sp7^+ cells led to reduced trabecular and cortical bones (*Figure 4b*, *Supplementary file 1*, *Source data 1*) and Sp7 expression (*Figure 4c*). Reduction in bone mass was more severe in the *Prrx1-Cre*-driven mutants compared to the *Sp7-Cre*-driven ones (*Figure 2d* and *Supplementary file 1*), suggesting that Piezo1/2 in Sp7^- cells such as the muscle, tenocytes and chondrocytes may also promote to bone formation. These are consistent with previous studies showing muscle contraction promotes bone formation (*Nowlan et al., 2010a*). In addition, we found that when cultured under osteogenic conditions, mechanical stress slightly increased *Piezo1* expression in the *Prrx1* lineage cells, not the *Sp7* lineage cells (*Figure 4d*). To further determine whether Piezo1/2 mediate the effects of mechanical stress in embryonic bone development, we decided to reduce both intrinsic and extrinsic mechanical forces by culturing mouse developing limb bud explants under static conditions, in which intact limb buds can grow and develop for several days (*Storm and Kingsley, 1999*; *Smith et al., 2013*; *Neubert et al., 1974*; *Decker et al., 2014*; *Figure 5a,b*, *Figure 5—source data 1*). We exploited this explant culture system to further interrogate the functions of Piezo1/2. Forelimb bud pairs were isolated from the left and right sides of E13.5 mouse embryos and treated for 4 days with PBS or Yoda, the only one molecule found to chemically regulate the opening of Piezo1 so far (*Syeda et al., 2015*), respectively, to minimize possible developmental variability from embryo to embryo. Bone development was slower in explant culture and there was no trabecular bone formation (*Figure 5a,b*). However, Yoda1 promoted bone formation and osteoblast differentiation in the wild type, but not the *Piezo1/2* DKO limbs (*Figure 5a*), suggesting that Piezo1 was functional in the explant culture. Importantly, the drastic difference in bone formation between the wild-type control and the *Sp7-Cre; Piezo1/2* DKO limbs developed in utero was eliminated (compare *Figure 5b* with *Figure 4a,c*). We next asked whether in adult life, bone loss under unloading conditions is caused by lack of Piezo1/2 stimulation. We adopted a mouse unloading model of Botulinum Toxin A (BTX)-induced hind limb muscle paralysis, which has gained prominence for its direct clinical relevance to immobilized patients, due to upper or lower motor neuron damage, muscular dystrophies or therapeutic bed rest (*Morse et al., 2014*; *Velders et al., 2008*; *Warner et al., 2006*). It has been shown that BTX injection causes loss of bone density, microstructure, and strength in both mice and rats. We therefore injected BTX into both the right quadriceps and the right calf muscles of 12 weeks old male mice. Left tibiae served as normal loading controls with only PBS injection to the same muscles. As a rapid and profound bone loss could be observed 1 week after BTX injection in male mice (*Grimston et al., 2007*), the mice were euthanized 10 days after BTX injection, and the tibiae were analyzed by μCT as shown previously (*Judex et al., 2004*). While BTX injection reduced bone mass in the wild-type mice, in the *Sp7-Cre*-driven *Piezo1/2* DKO mice, BTX injection did not cause significant bone loss, although the bone mass was lower compared to that of the wild-type controls (*Figure 5c*, *Figure 5—source data 1*). Taken together, Piezo1/2 are required for embryonic bone development and adult bone mass maintenance by mechanical forces.

## Yap1 and β-catenin activities were reduced in the *Piezo1* CKO and *Piezo1/2* DKO mutant bones

Severe bone reduction observed in the *Piezo1* or *Piezo1/2* mutants provided an unprecedented opportunity for us to identify the signaling pathways in bone development that are regulated by

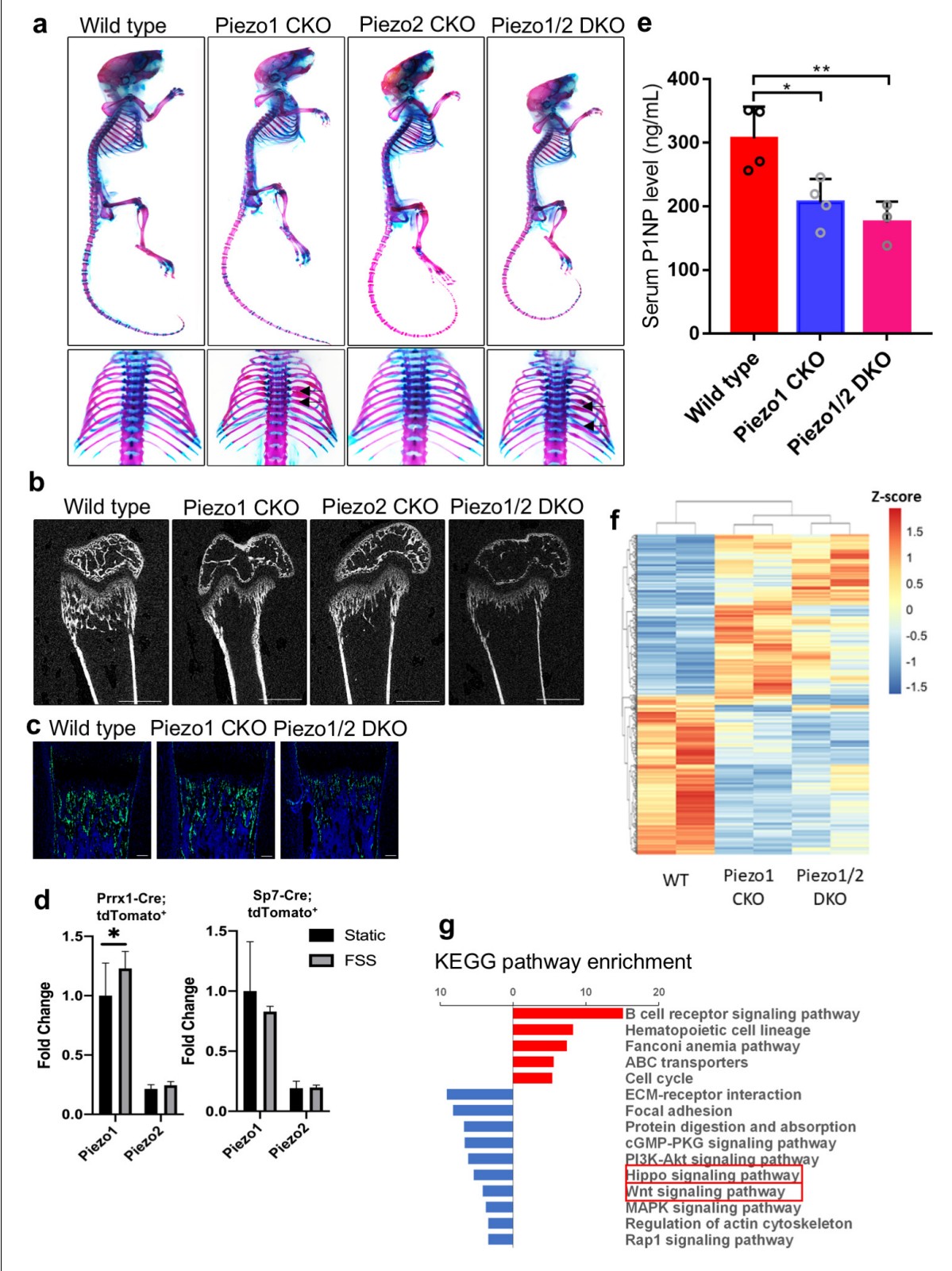

**Figure 4.** Loss of *Piezo1* or *Piezo1/2* in osteoblastic cells driven by the *Sp7-GFP::Cre* shows reduced bone mass and rib fractures. (a) Whole mount alizarin red and alcian blue staining of P21 mice with indicated genotypes. Dorsal view of the ribcages was shown in the lower panel. Arrows indicate bone fractures in the ribs. (b) Representative 2D μCT images of the femurs from P21 mice with indicated genotypes. The quantified parameters are shown in *Supplementary file 1*. (c) Expression of Sp7-Cre::GFP in humerus sections of P0 pups with indicated genotypes. (d) tdTomato+ cells were

*Figure 4 continued on next page*

*Figure 4 continued*

sorted from long bones of the *Prrx1-Cre;tdTomato^fl/+* and *Sp7-Cre;tdTomato^fl/+* mice and cultured under osteogenic differentiation under static or fluid shear stress (FSS) condition. *Piezo1* and *Piezo2* expression were analyzed using RT-qPCR. (e) Serum PINP levels in 6 weeks old mice of indicated genotypes detected by ELISA. As all bones were affected by the *Sp7-GFP::Cre* driver, PINP levels were significantly reduced in both *Piezo1* or *Piezo1/2* mutants (n = 3–4, means ± SD). (f) RNA samples from the P0 humerus bone tissues of indicated *Prrx1-Cre*-driven mutants were subject to RNA seq. Heat-map analysis of differentially expressed genes with fold change increase >2.82 fold and reduction to <0.35 fold. *Piezo1* CKO or *Piezo1/2* DKO mutants showed similar alteration of gene expression genome wide. (g) KEGG pathway analysis of differentially expressed genes. Prominent reductions in Hippo and Wnt signaling were identified (boxed). *p<0.05, **p<0.01, ***p<0.001, one-way ANOVA followed by Tukey's multiple comparisons tests when ANOVA was significant (*Figure 4—source data 1*).

The online version of this article includes the following source data and figure supplement(s) for figure 4:

**Source data 1.** Differentially expressed genes and orginal numbers for quantification.

**Figure supplement 1.** Gene ontology (GO) biological process analysis of differentially expressed genes.

Piezo1/2 channels gated by mechanical forces. We therefore isolated the developing humerus and femur bone tissues from the P0 wild type, *Prrx1-Cre* driven *Piezo1* CKO and *Piezo1/2* DKO pups and genome-wide gene expression was examined by performing RNA sequencing. The up- and down-regulated genes were subject to Gene Ontology (GO) enrichment analysis and KEGG pathway mapping (*Figure 4f*, *Figure 4—figure supplement 1*). We found that changes in reduced gene expression were enriched for osteoblast differentiation in the GO analysis and the Wnt/β-catenin (Ctnnb1) and Hippo/Yap1 signaling pathways in the KEGG pathway mapping. Wnt/Ctnnb1 signaling is essential for osteoblast differentiation (*Day et al., 2005*; *Hill et al., 2005*; *Hu et al., 2005*; *Glass et al., 2005*; *Gong et al., 2001*), and Yap1 and Wwtr1 are essential effectors of mechanotransduction to regulate cell proliferation and differentiation (*Halder et al., 2012*; *Aragona et al., 2013*; *Codelia et al., 2014*; *Wada et al., 2011*; *Dupont et al., 2011*). Loss of Yap1/Wwtr1 activities inhibits osteoblast differentiation in bone formation (*Kegelman et al., 2018*; *Pan et al., 2018*). We therefore focused our analyses on Wnt/Ctnnb1 and Hippo/Yap1 signaling. We found that Ctnnb1 and Yap1/Wwtr1 protein levels were both reduced in the bones of *Prrx1-Cre*-driven *Piezo1* CKO and *Piezo1/2* DKO mutants at P0 and more reduction was found in the *Piezo 1/2* DKO bone samples (*Figure 6a, b*, *Figure 6—source data 1*). In addition, by qRT-PCR analysis, we found that expression of transcriptional targets of both Yap1/Wwtr1 and Wnt/Ctnnb1 such as *Ctgf (Ccn2)/Cyr61(Ccn1)* and *Tcf7/Lef1*, respectively, were down-regulated in the *Piezo1* CKO and *Piezo1/2* DKO mutant bones (*Figure 6c*, *Figure 6—source data 1*). These results were further confirmed by in situ hybridization with a *Ccn2* probe using the RNAscope technology (*Figure 6d*). *Ccn2* expression was reduced in the developing humerus at E16.5 by removing *Piezo1/2*. These results show that Piezo1/2 activities may promote osteoblast differentiation by upregulating both Wnt/Ctnnb1 and Yap1 activities.

Mesenchymal stem cells (MSCs) from both the periosteum and bone marrow, give rise to osteoblasts (*Bianco et al., 2010*; *Schipani and Kronenberg, 2008*). We therefore determined whether BMSCs where MSCs are found also employ Piezo1/2 to regulate their osteoblast differentiation. Mammalian cells are surrounded by a mechanical microenvironment with neighboring cells and characteristic ECM, which provide mechanical cues that influence diverse biological processes including cell fate decisions (*Halder et al., 2012*; *Humphrey et al., 2014*). It is known that stem cells or progenitor cells differentiate optimally into distinct cell types when cultured at ECM elasticities that match the physiological ECM stiffness of their corresponding natural niche. Specifically, it was found that MSCs preferentially differentiate into osteoblasts when cultured on stiff surface in a rigid ECM environment, instead of adipocytes when cultured in a soft ECM environment (*McBeath et al., 2004*; *Kilian et al., 2010*). The stiff surface of regular tissue culture dishes supports osteoblast differentiation of BMSCs, which expressed both *Piezo1/2*, but *Piezo1* expression was stronger (*Figure 6—figure supplement 1a*). However, loss of *Piezo1* or *Piezo1/2* by Cre Adenovirus (Ad-Cre) infection of the *Piezo1^f/f* or *Piezo1^f/f; Piezo2^f/f* BMSCs (*Figure 6—figure supplement 1a*), respectively, led to reduced osteoblast differentiation and reduced Yap1/Wwtr1 and Wnt/Ctnnb1 signaling activities (*Figure 6e*, *Figure 6—source data 1*). Consistent with this, treating wild-type BMSCs with Yoda1 increased osteoblast differentiation as well as Wnt/Ctnnb1 and Yap1 transcriptional activities (*Figure 6—figure supplement 1b*, *Figure 6—figure supplement 1—source data 1*). Such osteogenic activity of Yoda1 was lost in the absence of *Piezo1/2* (*Figure 6f*, *Figure 6—source data 1*), suggesting that Yoda1 activity in promoting osteoblast differentiation depends on Piezo1. In addition,

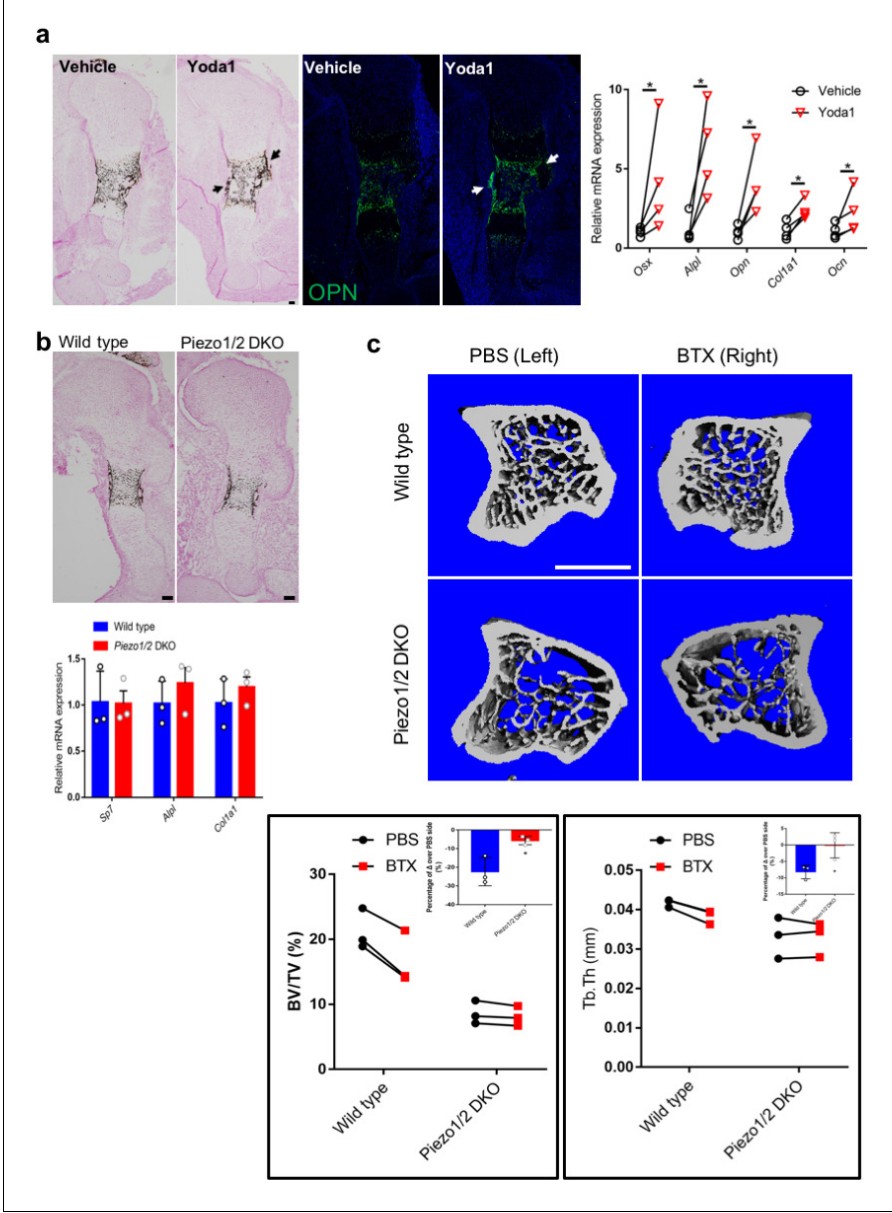

**Figure 5.** Loss of *Piezo1/2* diminished reduction in bone formation caused by unloading in both embryonic development and adult life. (a, b) Sections of the embryonic humerus from the limb bud dissected from E13.5 embryos and cultured for 4 days in BGJb medium under static conditions. Osteoblast differentiation was further determined by qRT-PCR anlysis of osteoblastic markers. (a) The right limb bud was treated with Yoda1 (400 nM), and the contralateral left limb bud was treated with equal volume of vehicle as control. Representative images of von Kossa or SPP1 staining of wild-type humerus from the cultured limbs. Scale bar: 100 μm. difference in osteoblast gene expression between limb bud pairs from the same embryo was shown on the right side. *p<0.05, according to paired ratio t test. (b) Representative images of von Kossa stained wild type or *Sp7-Cre*-driven *Piezo1/2 DKO* sections of the cultured humerus. Gene expression was shown below. No significant difference was found between the cultured wild type and *Piezo1/2* DKO limbs. (c) Representative μCT images of the tibia metaphysis in an unloading model of BTX-induced muscle paralysis. The BTX-injected right leg and the PBS-injected contralateral left leg from male mice are shown. Scale Bar: 1 mm. The BV/TV and trabecular bone thickness of each inject mouse were analyzed and compared. *p<0.05, according to paired ratio t test. The percentage of difference between BTX and PBS injected side over the PBS injection side was calculated and shown in the inserts. *p<0.05, according to unpaired student t test (*Figure 5—source data 1*).
The online version of this article includes the following source data for figure 5:

**Source data 1.** Original numbers collected for quantification.

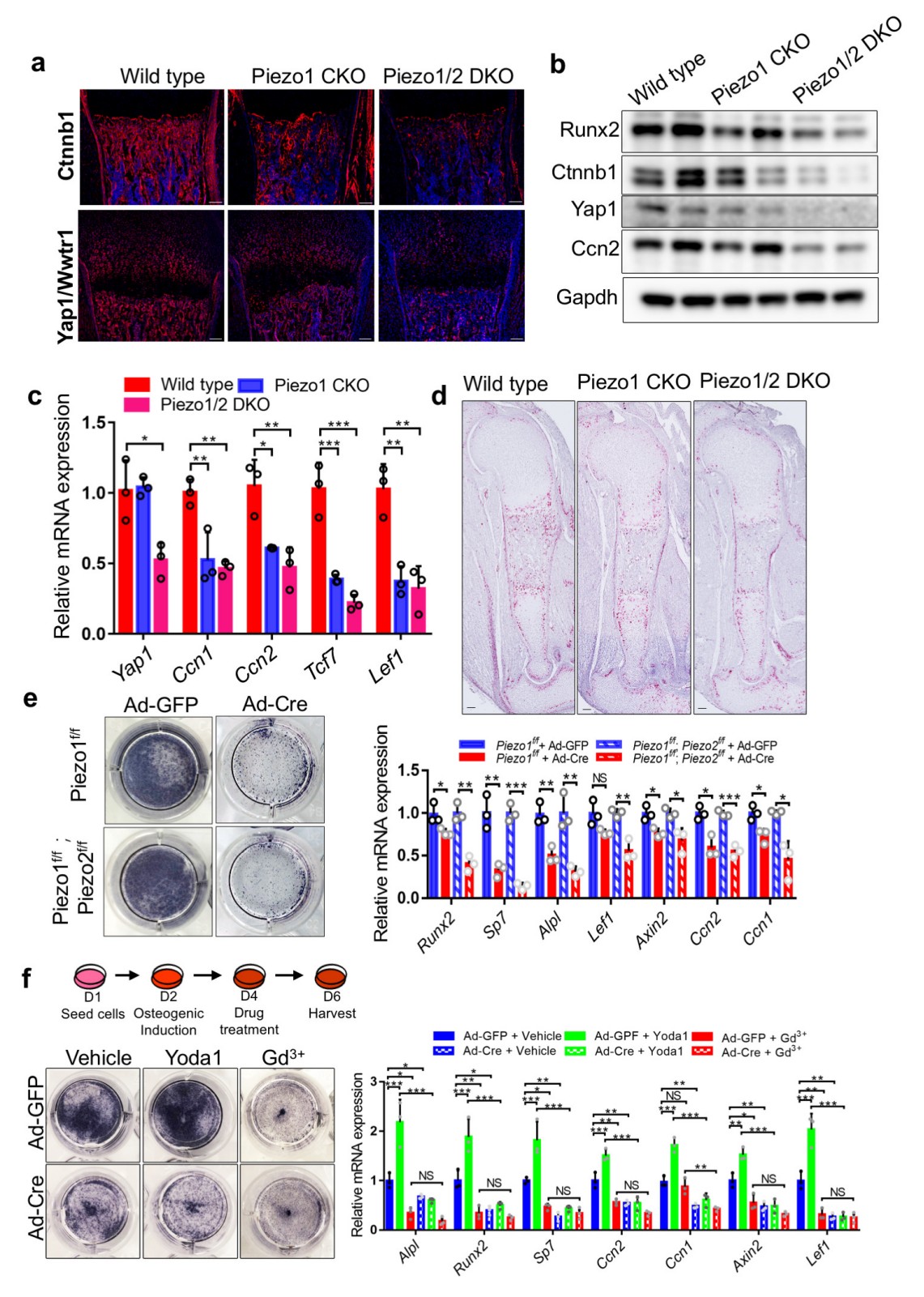

**Figure 6.** Loss of *Piezo1/2* reduced Yap1 and Wnt/Ctnnb1 signaling activities and osteoblast differentiation in vivo and in vitro. (**a**) Immunofluorescent staining of Ctnnb1 and Yap1/Wwtr1 in the humerus sections from the P0 *Prrx1-Cre* driven *Piezo1 and Piezo1/2* mutants and littermate controls. (**b**) Western blot analyses of P0 femur bone tissue lysates from P0 pups. (**c**) qPCR analyses of gene expression from femur bone tissues of P0 pups (n = 3, means ± SD). (**d**) *Ccn2* expression by RNAScope analysis of humerus sections of E16.5 embryos. (**e**) Alkaline phosphatase staining (left) and qPCR
*Figure 6 continued on next page*

*Figure 6 continued*

analyses (right, n = 3, means ± SD) in the indicated BMSCs infected with Ad-GFP or Ad-Cre 5 days after osteogenic induction. (f) Alkaline phosphatase staining (left) and qPCR analyses (right, n = 3, means ± SD) in *Piezo1^{f/f};Piezo2^{f/f}* BMSCs treated with Piezo1 agonist (Yoda1) and antagonist (Gd^{3+}) 6 days after osteogenic induction. The schematics of the induction process is shown on the top. All scale bars: 100 μm. DAPI (blue) stained the nucleus. *p<0.05, **p<0.01, ***p<0.001, two-tailed unpaired Student's *t*-test (e) or one-way ANOVA followed by Tukey's multiple comparisons tests when ANOVA was significant (c and f) (*Figure 6—source data 1*).

The online version of this article includes the following source data and figure supplement(s) for figure 6:

**Source data 1.** Original numbers and Western blots.
**Figure supplement 1.** Regulation of NFAT and CTNNB1 signaling by Piezo.
**Figure supplement 1—source data 1.** Original numbers and Western blots.

blocking Ca^{2+} influx with Gadolinium (Gd^{3+}), a potent blocker of Calcium channels including the Piezos (*Coste et al., 2010*), blocked osteoblast differentiation similar to loss of *Piezo1/2* (*Figure 6f*). These results suggest that Piezo1/2 depend on Ca^{2+} influx and mediate mechanotransduction via the Wnt/Ctnnb1 and Yap1 pathways in regulating osteoblast differentiation of BMSCs.

## Piezo1/2 sense both FSS and matrix rigidity

It is known that BMSCs and osteoblast lineage cells senses both FSS and ECM rigidity, which play critical roles in the fate determination of BMSCs reviewed by *Vining and Mooney (2017)*. A role for FSS as regulator of cell function has been recognized in different biological systems including the canalicula in bones (*Burger et al., 1995*), where fluid flow is physiologically present. As a proof of concept study to test the roles of Piezo1 and Piezo2 in mechanotransduction in BMSCs, we first applied biomechanical stimulation via a simple rocking platform to monolayer BMSC cultures in vitro (*Delaine-Smith et al., 2012*; *Wittkowske et al., 2016*; *Zhou et al., 2010*). The rocking 'see-saw' systems generate oscillatory FSS (*Zhou et al., 2010*), which is capable of directing BMSCs toward osteogenic differentiation by increasing alkaline phosphatase (Alpl) activity and deposition of mineralized matrix (*Delaine-Smith et al., 2012*; *Wittkowske et al., 2016*). We found that FSS stimulated osteoblast differentiation of wild-type BMSCs while loss of *Piezo1/2* completely abolished the response of BMSC to FSS in the osteogenic assay (*Figure 7a*, *Figure 7—source data 1*). Importantly, FSS upregulated protein levels of Ctnnb1 and Yap1, while levels of phosphorylated Ctnnb1 and Yap1, which are cytoplasmic and destined for degradation, were reduced by FSS in BMSCs, indicating that FSS activated both Ctnnb1 and Yap1 activities (*Figure 7b*). *Piezo1/2* loss in BMSCs abolished osteogenic response to FSS, reduced Ctnnb1 and Yap1 protein levels and increased Ctnnb1 and Yap1 phosphorylation (*Figure 7a,b*). Interestingly, increased Yap1 phosphorylation was not accompanied by increase of phosphorylated and activated Stk4 (Mst1) or Lats1 kinases (*Figure 7b*), suggesting that FSS-stimulated Yap1 activation may be independent of the Stk3/4 and Lats kinase cascade that controls Yap1 phosphorylation in canonical Hippo signaling (*Wu et al., 2003*; *Huang et al., 2005*; *Edgar, 2006*; *Pan, 2007*). These results suggest that Piezo1/2 are key mechanical force sensors that relay FSS signals to promote osteoblast differentiation through Yap1 and Ctnnb1 signaling.

As Piezo1/2 are Calcium (Ca^{2+}) permeable channels and Ca^{2+} signaling plays crucial roles in bone biology (*Zayzafoon, 2006*), we then determined whether Piezo1/2 regulate osteoblast differentiation through Ca^{2+} influx. To determine whether *Piezo1/2* are required for mechanical force-induced Ca^{2+} influx in the BMSCs, we employed the *PC::G5-tdT* mouse strain, which expresses the green fluorescent calcium indicator protein GCaMP5G and the tdTomato red fluorescent protein when induced by Cre recombinase (*Gee et al., 2014*). The basal level Ca^{2+} signal was stronger in the wild-type BMSCs compared with the *Piezo1/2* deficient BMSCs (*Figure 6c*, *Figure 6—source data 1*). More importantly, when BMSCs were subject to FSS, more robust Ca^{2+} influx was observed in the wild-type BMSCs compared to the *Piezo*-deficient ones (*Figure 7c* and *Video 1*). Furthermore, Yoda1 treatment led to more robust Ca^{2+} influx in the wild type, but not Ad-Cre induced *Piezo1/2*-deficient BMSCs (*Figure 7—figure supplement 1*, *Video 2*, *Figure 7—figure supplement 1—source data 1*). These results together indicate that Piezo1/2 channels in BMSCs sense FSS-like mechanical forces to allow Ca^{2+} influx, leading to osteoblast differentiation.

We next determined whether Piezo1/2 also sense mechanical stress induced by ECM rigidity, which is another major mechanical cue that promotes osteoblast differentiation and Yap1 activation

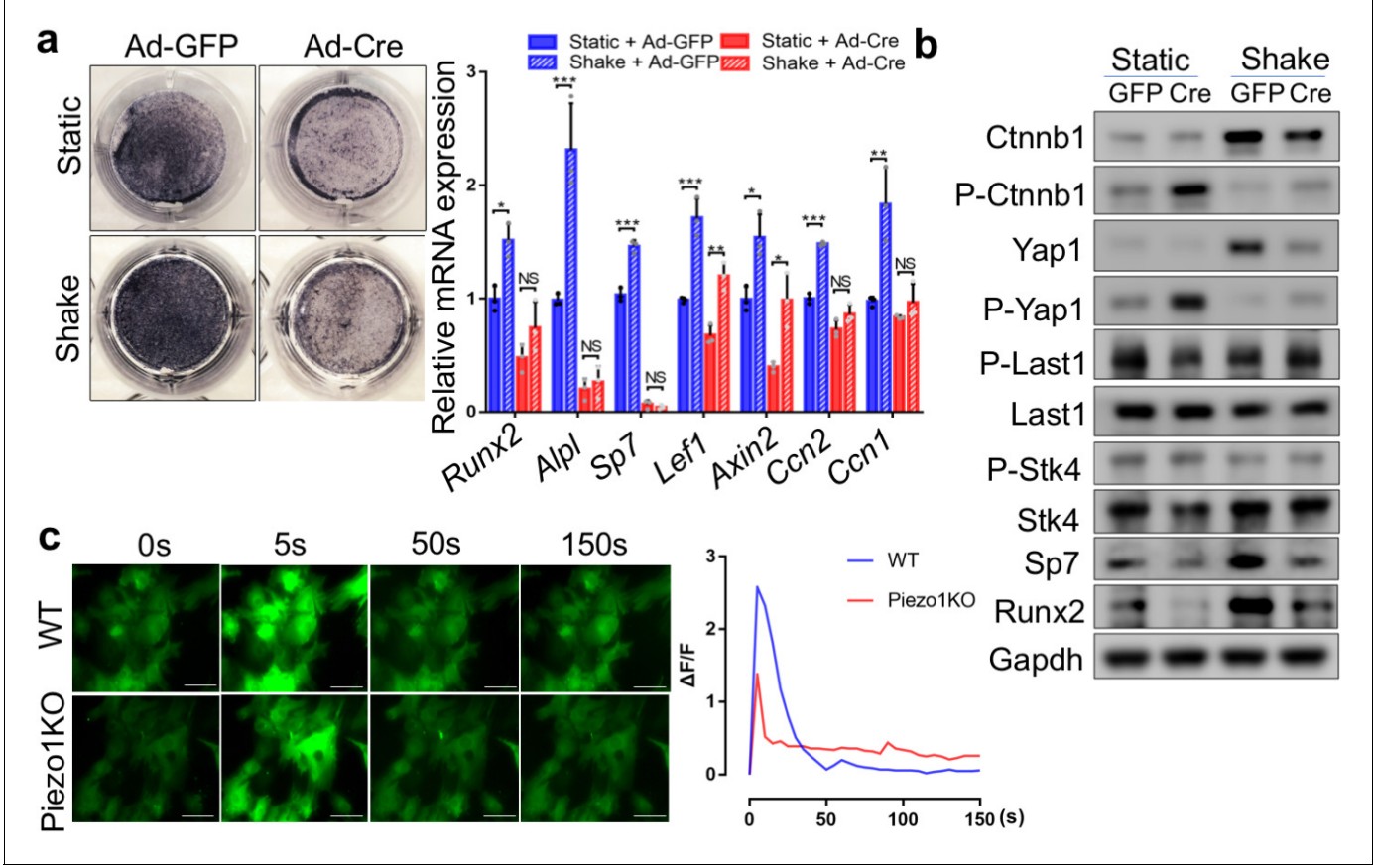

**Figure 7.** Piezo1 was required to sense mechanical forces generated by FSS and upregulate Yap1 and Ctnnb1 activities in primary mouse BMSCs. (a) Alkaline phosphatase staining (left) and qPCR analyses (right) in BMSCs cultured with or without shaking during osteogenic induction (n = 3, means ± SD). *p<0.05, **p<0.01, ***p<0.001, one-way ANOVA followed by Tukey's multiple comparisons tests (*Figure 7—source data 1*). (b) Western blotting analyses of cell lysates from BMSCs cultured under static or shaking conditions during osteogenic induction. (c) Representative images (left) of BMSCs with fluorescent GCaMP5G reporter at different time points after FSS. Scale bars: 80 μm. The fluorescent intensities of GCaMP5G were quantified (right) every 5 s (*Figure 7—source data 1*).

The online version of this article includes the following source data and figure supplement(s) for figure 7:

**Source data 1.** Original Western blots and numbers collected for quantification.

**Figure supplement 1.** Representative GFP images of the GCaMP5G Ca$^{2+}$ reporter in Ad-Cre infected BMSCs at different time points after Yoda1 treatment.

**Figure supplement 1—source data 1.** Original numbers collected for quantification.

(*Halder et al., 2012*; *Dupont et al., 2011*; *McBeath et al., 2004*; *Meng et al., 2018*). It is known that the plasma membrane stretches differentially when plated in matrices with different stiffness and cell spreading strongly correlates with matrix stiffness, osteoblast differentiation and Yap1 nuclear localization (*Halder et al., 2012*; *Dupont et al., 2011*; *Meng et al., 2018*; *Sunyer et al., 2012*; *Eyckmans et al., 2012*). Using this system, we plated BMSCs on soft (1 kPa) or stiff (40 kPa) hydrogels to test whether Piezo1/2 are also required to sense matrix stiffness. While the wild-type BMSC have difficulty spreading on a soft matrix (1 kPa) with diffused Yap1 localization in both cytoplasmic and nuclear compartments, on a stiff matrix (40 kPa), they spread to much larger areas with strong nuclear Yap1 localization (*Figure 8a*). Importantly, the *Piezo1/2*-deficient BMSCs plated on a stiff matrix behaved more like the wild-type BMSCs plated on a soft matrix. They showed weaker intracellular Ca$^{2+}$ signaling (*Figure 8—figure supplement 1*, *Figure 8—figure supplement 1— source data 1*), were less spreading, contained less defined Yap1 nuclear localization (*Figure 8a*), and exhibited reduced Yap1 activities and osteoblast differentiation (*Figure 8b*). Osteogenic differentiation of BMSCs culture on the hydrogel was determined by ALP staining and quantifies as previously described (*Figure 8b*; *Dupont et al., 2011*). These results indicate that Piezo1/2 also promote

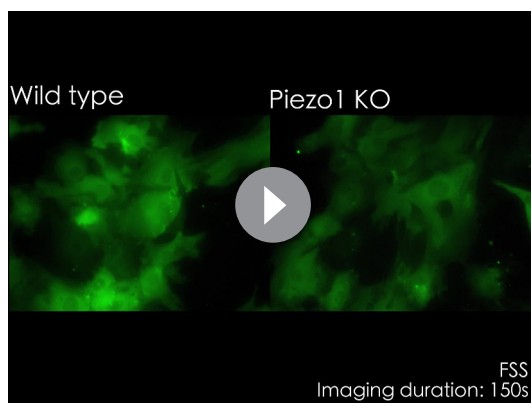

**Video 1.** Pseudovideo recording of the GFP signals of the intracellular Ca$^{2+}$ sensor GCaMP5G after FSS stimulation. Images were continuously taken at time intervals of 5 s, and this pseudovideo was made from the first parallel 30 images. Left: Ad-Cre virus infected primary BMSCs from the control *PC::G5-tdT* mice; Right: Ad-Cre virus infected primary BMSCs from the *Piezo1fl/fl*; *PC::G5-tdT* mice.

https://elifesciences.org/articles/52779#video1

osteoblast differentiation by sensing matrix stiffness that is known to regulates Yap1 nuclear localization.

To determine whether Piezo1/2-dependent Ca$^{2+}$ signaling regulates Yap1 and Ctnnb1 activities, we turned our attention to the calcium and calmodulin-dependent heterodimeric serine/threonine phosphatase calcineurin (Ppp3ca), which is a critical intracellular Ca$^{2+}$ sensor. To test whether Ppp3ca is required in the Piezo1/2 initiated mechanotransduction pathway to promote osteoblast differentiation, we treated the mouse primary BMSCs with a commonly used Ppp3ca inhibitor, Cyclosporin A (CsA) (*Liu et al., 1991*). CsA treatment abolished osteoblast differentiation as well as Yap1 and Ctnnb1 activation enhanced by FSS or Yoda1 treatment (*Figure 8c, d*, *Figure 8—source data 1*, *Figure 6—figure supplement 1b*, *Figure 6—figure supplement 1—source data 1*). These results suggest that Ppp3ca may act downstream of Piezo1/2 in mechanotransduction to regulate Yap1 and Ctnnb1 activities during osteoblast differentiation.

## Piezo1/2 act through Ppp3ca to regulate concerted activation of NFATc1, YAP1 and CTNNB1

Transcription factors in the nuclear factor of activated T cells (NFAT) family are major Ppp3ca substrates that regulate gene expression (*Li et al., 2011*; *Hogan et al., 2003*). We next investigated the roles of Ppp3ca and Nfatc1 in mediating Piezo1/2 regulation of mechanotransduction, as expression of constitutively nuclear NFATc1 variant (NFATc1$^{nuc}$) in osteoblast leads to high bone mass (*Winslow et al., 2006*) and loss of Nfatc1/2 led to reduced osteoblast differentiation and bone for-

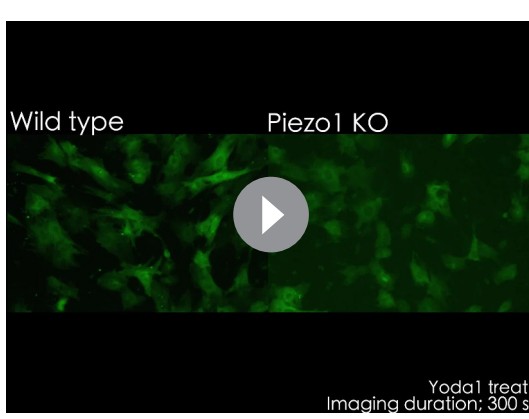

**Video 2.** Pseudovideo recording of the GFP signals of the intracellular Ca$^{2+}$ sensor GCaMP5G after Yoda1 treatment. Images were continuously taken at time intervals of 5 s, and this pseudovideo was made from the first parallel 60 images. Left: Ad-Cre virus infected primary BMSCs from the control *PC::G5-tdT* mice; Right: Ad-Cre virus infected primary BMSCs from the *Piezo1fl/fl*; *PC::G5-tdT* mice.

https://elifesciences.org/articles/52779#video2

mation (*Winslow et al., 2006*; *Koga et al., 2005*). We found that loss of *Piezo1/2* together, led to a strong increase of Nfatc1 phosphorylation in the developing bone (*Figure 9a*, *Figure 9—source data 1*). Loss of *Piezo1* alone also increased Nfatc1 phosphorylation, although to a lesser extent (*Figure 9a*). As Nfat transcription factors are highly phosphorylated and cytoplasmic in unstimulated cells, and an increase in intracellular Ca$^{2+}$ leads to Nfatc1 dephosphorylation by Ppp3ca and its subsequent nuclear translocation to activate downstream gene expression (*Liu et al., 1997*; *Masuda et al., 1997*; *Beals et al., 1997*; *Crabtree and Olson, 2002*), these results suggest that Piezo1/2 activate Nfatc1 by inhibiting its phosphorylation in the developing bone. To further determine whether PIEZO1/2 regulate NFATc1 transcriptional activities, we first tested whether activation of PIEZO1 by Yoda1 alters transcriptional activities of NFATc1 in a luciferase assay in the HEK 293 T cells, in which endogenous PIEZO1 is expressed and mediates mechanosensation (*Dubin et al., 2017*; *Bae et al., 2013*; *Figure 9b*,

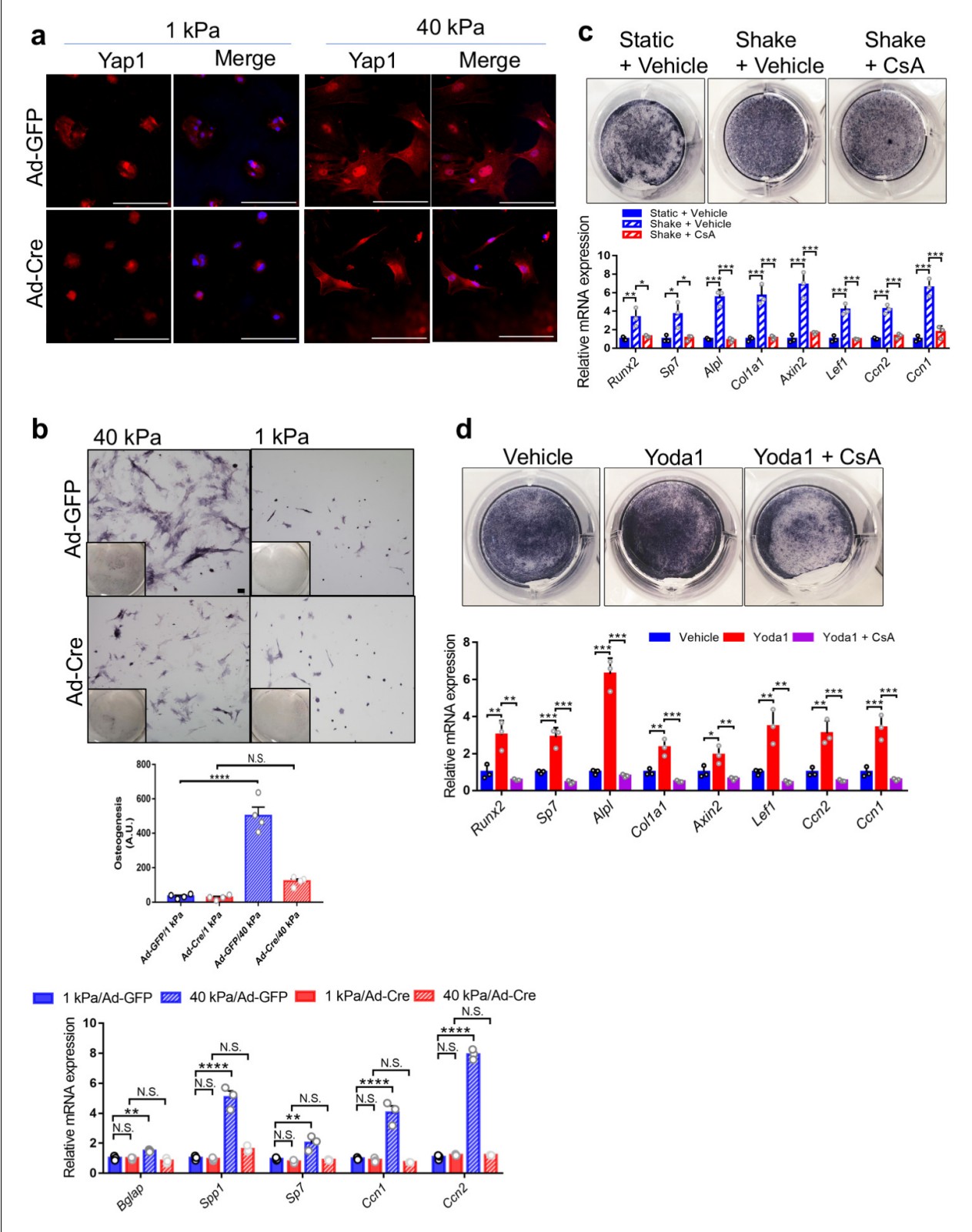

**Figure 8.** Piezo1 was required to sense ECM stiffness and upregulate Yap1 and Ctnnb1 activities in primary mouse BMSCs. (a) Yap1 immunostaining of *Piezo1^{f/f};Piezo2^{f/f}* BMSCs seeded on matrices with the indicated stiffness after Ad-GFP or Ad-Cre infection. Scale bars: 100 μm. (b) BMSC were infected with the indicated Ad-virus, plated on stiff (40 kPa) or soft (1 kPa) substrates and induced to differentiate into osteoblasts for 6 days. Representative alkaline phosphatase stainings images were shown. Scale bar: 100 mm. As shown previously (*Dupont et al., 2011*). Osteogenic differentiation was

*Figure 8 continued on next page*

*Figure 8 continued*

quantified by the alkaline-phosphatase-positive area determined with ImageJ as the number of blue pixels across the picture. This value was normalized to the number of cells (Hoechst/nuclei) for each picture (arbitrary units). RT-PCR anlaysis of BMSCs grown on the indicated hydrogels was shown below. (c) Alkaline phosphatase staining and qPCR analyses of primary mouse BMSCs treated with shaking with or without a Ppp3ca inhibitor CsA (n = 3, means ± SD). (d) Alkaline phosphatase staining and qPCR analyses of BMSCs treated with Yoda1 and CsA (n = 3, means ± SD). *p<0.05, **p<0.01, ***p<0.001, one-way ANOVA followed by Tukey's multiple comparisons tests (b–d) (*Figure 8—source data 1*).

The online version of this article includes the following source data and figure supplement(s) for figure 8:

**Source data 1.** Original numbers for quantification.
**Figure supplement 1.** *Piezo* mutant primary BMSCs were defective in mechanotransduction.
**Figure supplement 1—source data 1.** Original Western blots.

*Figure 9—source data 1*). Yoda1 treatment stimulated transcription activities of not only NFAT, but also YAP1 and CTNNB1, all of which were abolished by knocking down *PIEZO1* (*Figure 9b*). These results indicate that activation of PIEZO1 upregulates NFATc1, YAP1 or CTNNB1 transcriptional activities. As Ppp3ca inhibition by CsA treatment suppressed the cellular and molecular effects of FSS or Yoda1 treatment in BMSCs (*Figure 8c, d*), Nfatc1 activation by Ppp3ca likely mediated Piezo1-dependent Yap1 and Ctnnb1 activation. Indeed, transcriptional activities of NFAT, YAP1 and CTNNB1 were mutually enhanced, with the only exception that YAP1 did not seem to enhance NFAT activities in HEK293T cells (*Figure 9—figure supplement 1b*, *Figure 9—figure supplement 1—source data 1*). To further understand the mechanism underlying YAP1 and CTNNB1 activation in concert with NFATc1, we first tested YAP1 and NFATc1 interaction by co-immunoprecipitation (Co-IP) assays and found such interaction was promoted by FSS or Yoda1 treatment, but inhibited by CsA or $Gd^{3+}$ treatment (*Figure 9c*, *Figure 9—source data 1*, *Figure 9—figure supplement 1b*, *Figure 9—figure supplement 1—source data 1*). These results suggest that PIEZO1-induced $Ca^{2+}$ signaling promoted NFATc1 interaction with YAP1. To further confirm that PIEZO1 may activate YAP1 by activating PPP3CA and NFATc1, we examined NFATc1 and YAP1 nuclear localization in HEK293 cells and found that both were robustly promoted by FSS (*Figure 9d*). In addition, overexpression of the catalytic subunit of PPP3CA promoted YAP1 dephosphorylation (*Figure 9—figure supplement 1c*, *Figure 9—figure supplement 1—source data 1*). Moreover, Yoda1 treatment mimicked the effects of FSS treatment or PPP3CA expression (*Figure 9e*, *Figure 9—figure supplement 1c*). Blocking $Ca^{2+}$ influx with $Gd^{3+}$ treatment or PPP3CA activity with CsA treatment both inhibited nuclear localization of NFATc1 and YAP1 induced by FSS or Yoda1 treatment (*Figure 9e*). Furthermore, NFATc1 interaction with YAP1 or CTNNB1 was synergistically promoted by the other, as presence of all three transcription factors robustly increased their interactions (*Figure 9—figure supplement 1d*, *Figure 9—figure supplement 1—source data 1*). It is important to note that like NFATc1, YAP1 and CTNNB1 are also activated by dephosphorylation. We therefore tested whether the Piezo1-mediated mechanotransduction promotes osteoblast differentiation by a concerted dephosphorylation of endogenous Yap1 and Ctnnb1 in mouse primary BMSCs. Indeed, Yoda 1 treatment quickly reduced Yap1 and Ctnnb1 phosphorylation (*Figure 9f*). Such activity of Yoda1 was abolished in the *Piezo1/2*-deficient BMSCs (*Figure 9—figure supplement 1e*). Conversely, blocking $Ca^{2+}$ influx by $Gd^{3+}$ treatment promoted Yap1 and Ctnnb1 phosphorylation while reducing their protein levels within 30 min (*Figure 9—figure supplement 1f*). Furthermore, in the mouse primary BMSCs, Ppp3ca was found to bind not only Nfatc1, but also Yap1 and Ctnnb1. These interactions were enhanced by Yoda1 treatment and inhibited by CsA (*Figure 9g*, *Figure 9—source data 1*). Taken together, our data indicate that $Ca^{2+}$/Ppp3ca signaling activated by Piezo1 leads to concerted activation of Yap1 and Ctnnb1 with Nfat, all of which act together to regulate transcription changes that promote osteoblast differentiation and bone formation.

## Enhanced Ctnnb1 activities partially rescued the bone defects of *Piezo1/2* bone mutants

The well-established function of Wnt/Ctnnb1 signaling activities in both bone formation and maintenance led us to test whether reduced bone formation in the *Piezo1/2* mutants could be rescued by enhancing the Wnt/Ctnnb1 activities. We injected pregnant females and new born mouse pups with BIO, a potent specific inhibitor for GSK3, a kinase that promotes Ctnnb1 degradation (*Sato et al.,*

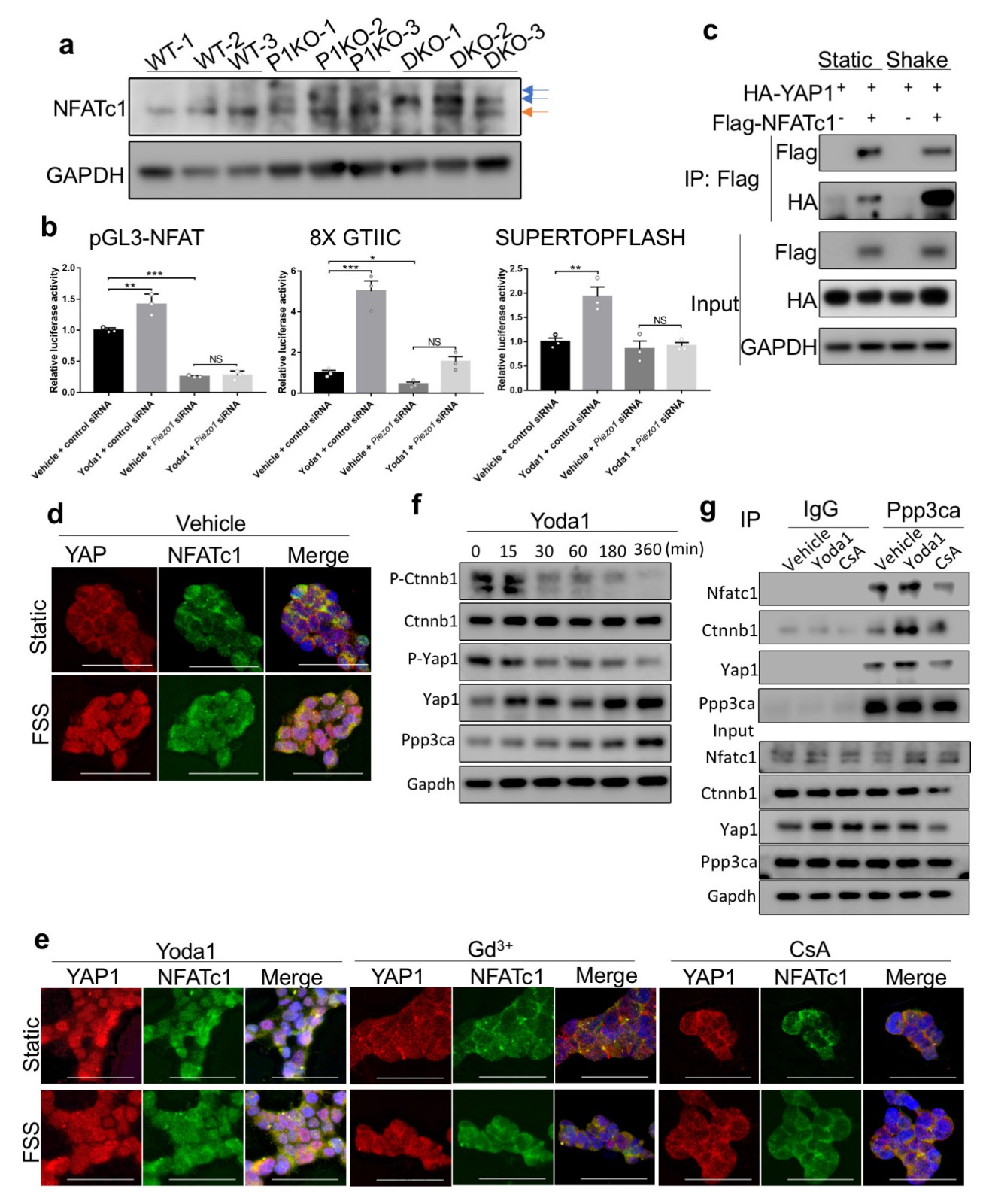

**Figure 9.** Concerted activation of NFATc1, Yap1 and Ctnnb1 by Piezo channel activation. (a) Western blotting analysis of NFATc1 expression in bone tissue lysates from P0 pups. The slower migrating phosphorylated NFATc1 forms were indicated by blue arrows. (b) Luciferase reporter assays of NFATc1 (left), YAP1 (middle) and Ctnnb1 (right) activities in HEK293T cells (n = 3, means ± SD). *p<0.05, **p<0.01, ***p<0.001, two-tailed unpaired Student's *t*-test and one-way ANOVA followed by Tukey's multiple comparisons tests when ANOVA was significant (*Figure 9—source data 1*). Yoda1

*Figure 9 continued on next page*

*Figure 9 continued*

treatment promoted transcription activities of NFATc1, YAP1 and Ctnnb1, which were abolished by PIEZO1 knocking down. (**c**) Immunoprecipitation (IP) assays of HEK293T cell lysates with the indicated expression constructs. Shaking promoted NFATc1 and YAP1 binding. (**d, e**) Immunostaining of YAP1 and NFATc1 in HEK293 cells with indicated treatments. Activation of PIEZO1 with Yoda1 or FSS through shaking promoted nuclear localization of both YAP1 and NFATc1, which was blocked by $Gd^{3+}$ or CsA treatment. Scale bars, 100 μm. (**f**) Western blotting analyses of Yoda1 treated primary mouse BMSCs. Phosphorylated Ctnnb1 and Yap1 were quickly reduced. (**g**) IP assays of Ppp3ca binding to NFATc1, Yap1 and Ctnnb1 in mouse primary BMSCs. IP with IgG was a negative control. Yoda1 treatment promoted CnA binding to NFATc1, Yap1 and Ctnnb1, which was inhibited by CsA.

The online version of this article includes the following source data and figure supplement(s) for figure 9:

**Source data 1.** Original Western blots and data for quantification.
**Figure supplement 1.** PIEZO1 activation led to concerted activation of NFATc1, YAP1 and CTNNB1.
**Figure supplement 1—source data 1.** Original Western blots.

*2004*; *Figure 10—figure supplement 1a*, *Figure 10—figure supplement 1*; *Source data 1*). We found that BIO treatment increased long bone length and bone mass with reduced bone fractures in the *Piezo1/2* DKO mice (*Figure 10a*, *Figure 10—source data 1*, *Figure 10—figure supplement 1a, b*). Further analyses showed that BIO treatment enhanced Sp7 and Spp1 expression while Ctsk expression was reduced (*Figure 10b*). Thus, increased Ctnnb1 activities enhanced osteoblast differentiation and maturation while osteoclast differentiation was inhibited. To test whether Ctnnb1 and Yap1 also enhances each other's activity in BMSCs, we treated the BMSCs with BIO or XMU-MP-1, an inhibitor of Hippo kinase Stk3/4 (*Fan et al., 2016*), and found that while BIO increased Ctnnb1 protein levels and reduced Ctnnb1 phosphorylation as expected, it also increased Yap1 protein levels and reduced Yap1 phosphorylation (*Figure 10—figure supplement 1c*). Likewise, reduction of Yap1 phosphorylation caused by XMU-MP-1 treatment was also accompanied by increase in total and non-phosphorylated Ctnnb1 protein levels. In addition, blocking Yap1 transcription activity by Verteporfin (VP) (*Liu-Chittenden et al., 2012*) reduced the effects of BIO in promoting both Ctnnb1 and Yap1 levels, suggesting that Ctnnb1 requires Yap1 activity to be fully upregulated (*Figure 10—figure supplement 1c*). Consistently, BIO treatment of BMSCs in vitro partially rescued reduction of osteoblast differentiation due to *Piezo1/2* loss (*Figure 10—figure supplement 1d*). However, while Ctnnb1 protein levels were increased by BIO treatment, the increase of Yap1 was less appreciable (*Figure 10c*), suggesting that while Ctnnb1 activation is essential for osteoblast differentiation and can promote Yap1 activation, Yap1 is also regulated by Piezo1/2 by Ctnnb1-independent pathways and it is the concerted activation of NFATc1, Yap1 and Ctnnb1 that constitutes a fully functional mechanotransduction pathway downstream of Piezo channels in promoting bone formation.

## Discussion

Here, we have identified the mechanically activated cation channels.

Piezo1 and 2 as essential mechanosensors that promote osteoblast differentiation in both embryonic development and adult bone homeostasis. We have also identified a Piezo1/2-initiated mechanotransduction pathway in bone development. While Piezo2 is dispensable for bone development, it shares redundant functions with Piezo1. We show that Piezo1 and 2 are required to sense FSS and matrix rigidity, two crucial mechanical cues that promote osteoblast differentiation and bone formation, through intracellular $Ca^{2+}$ signaling that regulates the concerted activation of NFAT, Yap1, and Ctnnb1 in controlling osteoblast cell fate. These findings fill a major knowledge gap in understanding how external mechanical forces, molecular force sensors, intracellular biological signaling pathways and nuclear transcriptional regulation drive final cellular outcomes are likely to act in skeleton and other tissues and organs. Human PIEZO1 single nucleotide polymorphisms (SNPs) are associated with body height reduction (*Marouli et al., 2017*) and may be osteoporosis based on a recent GWAS of UK Biobank (*Morris et al., 2019*). Piezo1/2 are also the first type of MS channels documented to underlie a human disease linked to mechanical pathologies including a number of blood disorders and problems with proprioception (*Martinac, 2012*; *Coste et al., 2012*; *Murthy et al., 2017*; *Chesler et al., 2016*). Therefore, our findings provide molecular insights into the integration between mechanical stimulation and biochemical signaling pathways in developmental and homeotic events in skeletal and other tissues and organs. It is interesting to note that our observations are

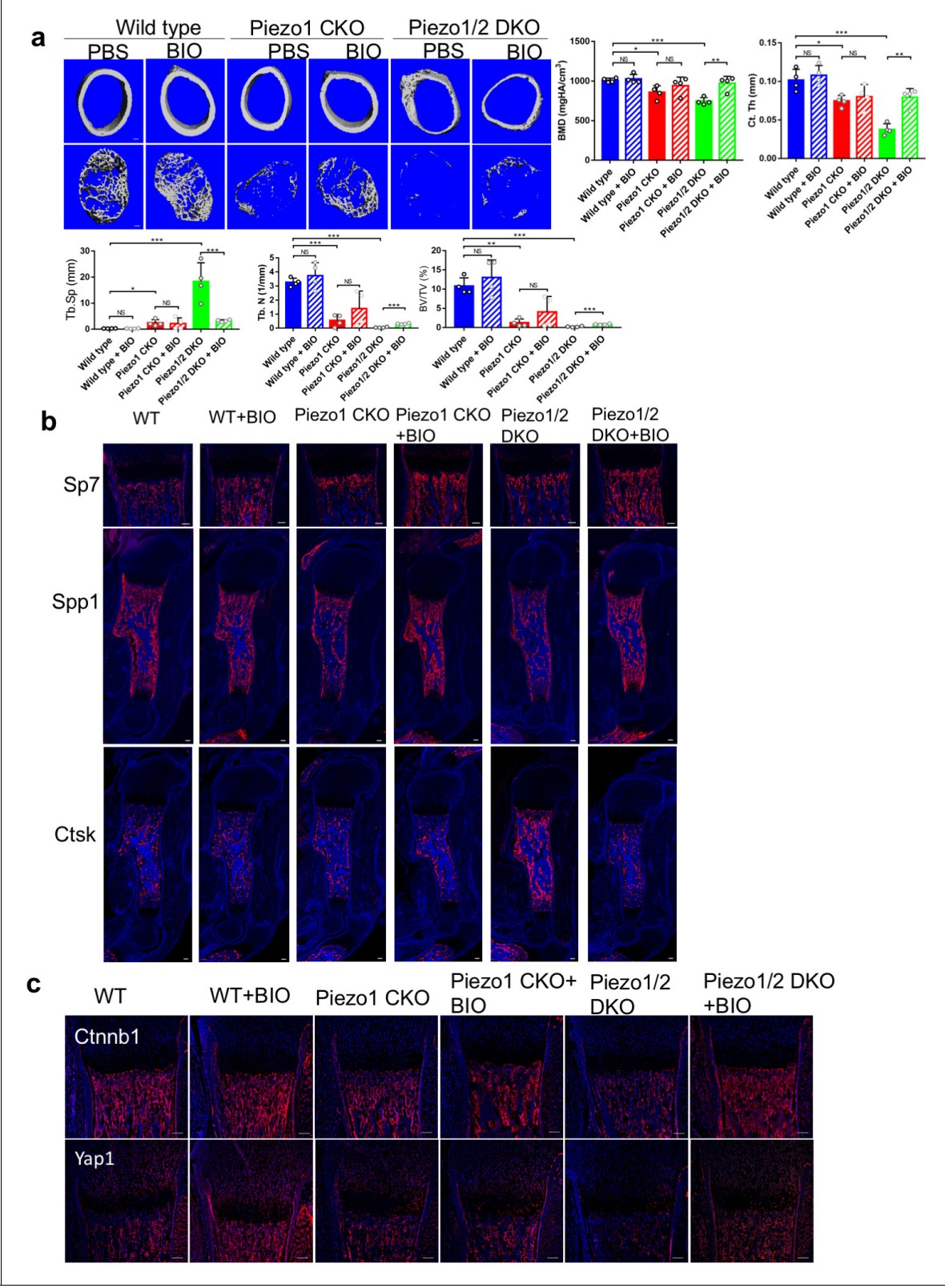

**Figure 10.** GSK3 inhibitor BIO partially rescued the bone development defects in the *Piezo1/2* mutants. (a) Representative cross section images and quantification of μCT scans of the femurs from P21 mice. N = 4. *p<0.05, **p<0.01, ***p<0.001, two-tailed unpaired Student's *t*-test and one-way ANOVA followed by Tukey's multiple comparisons tests when ANOVA was significant. Data are shown as means ± SD (*Figure 10—source data 1*). (b) *Figure 10 continued on next page*

*Figure 10 continued*

Immunostaining of humerus sections from P0 mouse pups. (**c**) Representative immunofluorecent images of Yap1 and Ctnnb1 staining of humerus sections from P0 mouse pups with the indicated genotypes. All scale bars: 100 µm. DAPI (blue) stained the nucleus.

The online version of this article includes the following source data and figure supplement(s) for figure 10:

**Source data 1.** Original data for quantification.
**Figure supplement 1.** BIO injection partially rescued the reduced bone mass phenotypes in *Piezo* mutants.

consistent and supported by recently published work showing that Piezo1 is required in osteoblasts and osteocytes for bone formation or resorption in postnatal mice (*Sun et al., 2019*; *Li et al., 2019*).

The Wolff's law developed in the 19th century, that bone adapts to pressure, or a lack of it reflects the physiological importance of mechanotransduction in bone health. Great strides have been made to understand the cellular responses to mechanical loading since then, but little is known about how mechanical stimuli are sensed and transduced at molecular levels. ECM-cell and cell-cell interactions are fundamentally important components of the mechanoenvironment essential for bone formation and function. Much of the past attention has been focused on biochemical interactions in the mechanoenvironment. In this regard, the integrin signaling pathway and the downstream actomyosin cytoskeleton have been extensively studied and recognized as major players in assessment of the mechanics, by which cells initiate a series of mechanoregulatory processes (*Humphrey et al., 2014*). Our genetic and biochemical studies of the Piezo1/2 channel-dependent mechanotransduction provide an additional mechanism for sensing the mechanoenvironment and initiation of mechanotransduction. Outside the bone, PIEZO1 activates integrin-FAK (Ptk2) signaling in glioblastoma cells (*Chen et al., 2018*) and vascular endothelial cells (*McHugh et al., 2012*; *Albarrán-Juárez et al., 2018*). However, Piezo1-dependent mechanotransduction in endothelial cells could also be integrin-independent (*Albarrán-Juárez et al., 2018*). As integrin signaling is required for osteoblast differentiation during bone formation (*Marie et al., 2014*) and *Piezo1/2* is required for cell spreading and Ptk2 activation in BMSCs (*Figure 8a,b*, *Figure 8—source data 1*, *Figure 8—figure supplement 1b,c*), it will be important to further examine integrin-independent and integrin-dependent Piezo1/2 activities in mechanotransduction of skeletal cells. It has been shown that PIEZO1 channels sense force directly transmitted through the bilayer and they are activated by bilayer tension in bleb membranes largely free of cytoskeleton gating at lower pressures (*Cox et al., 2016*). Therefore, it appears that removal of the cortical cytoskeleton and the mechanoprotection it provides sensitizes PIEZO1 channels. This raises the possibility that actomyosin cytoskeleton could also be part of a feedback adaptation process that dampens cellular responses to mechanical loading.

A major finding in this study is the Piezo1/2-dependent activation of Ppp3ca and concerted activation of the NFAT-Yap1-Ctnnb1 transcription factor network. Yap1 and Wwtr1 are essential effectors of mechanotransduced regulation of cell proliferation and differentiation (*Halder et al., 2012*; *Aragona et al., 2013*; *Codelia et al., 2014*; *Wada et al., 2011*; *Dupont et al., 2011*). The Wnt/Ctnnb1 signaling pathway plays critical roles in regulating osteoblast differentiation. Our findings here raise the possibility that Yap1/Wwtr1 could also be activated quickly by MS channels like Piezo1/2 in a noncanonical pathway, in addition to currently known Yap1 regulators including focal adhesion, actomyosin cytoskeleton and Hippo/Lats kinase cascade (*Panciera et al., 2017*). Furthermore, such regulation was executed together with Ctnnb1 activation, suggesting that Wnt/Ctnnb1 activity can also be activated by mechanotransduction via Yap1 and may mediate some of Yap1 activity in promoting bone formation. It is interesting to note that the AP-1 transcription factor (Fos–Jun dimers) is a major transcriptional partner of both NFAT (*Macián et al., 2001*) and YAP1/WWTR1 (*Zanconato et al., 2015*). Composite cis-regulatory elements containing NFAT:AP-1 or TEAD (the DNA binding partner of YAP1/WWTR1):AP-1 motifs are found in the regulatory regions of many target genes. Cooperative binding to these composite cis-regulatory elements have been found between NFAT or YAP1/WWTR1 with AP-1. Our findings here specifically advanced current understanding of Yap1 activation in mechanotransduction by connecting Piezo1/2 mediated $Ca^{2+}$ signaling with a signaling cascade leading to downstream actomyosin cytoskeleton rearrangement and a concerted activation of key transcription factors including Yap1, Ctnnb1 and NFAT, all of which have been demonstrated to play key roles in promoting osteoblast differentiation. In light of the NFAT

and Yap1 interactions we found here, it will be important to further test the role of AP-1 in NFAT interaction with YAP1; to find out whether cis-regulatory elements binding NFAT, AP-1 and TEAD in close vicinity form super enhancers and whether the genes they control are enriched in transcriptional responses to mechanical forces.

It is well established that fluid mechanical stimulation provides important mechanical stimuli to regulate bone development and regeneration in vitro and in vivo. FSS is known to acutely increase intracellular calcium concentrations in BMSCs, promote cell proliferation and upregulate osteogenic gene expression (*Li et al., 2004*). The important roles of intracellular calcium in osteoblast differentiation has been studied extensively (*Zayzafoon, 2006*; *Eapen et al., 2010*; *Ishikawa et al., 2011*; *Choi et al., 2018*). For instance, it is shown that Pannexin 3 functions as an ER $Ca^{2+}$ channel to promote osteoblast differentiation. Pharmacological reduction or increase of intracellular $Ca^{2+}$ concentration have been used to alter signaling pathways that regulate osteoblast differentiation. Our findings that Piezo1/2 regulate osteoblast differentiation by regulating Ppp3ca/NFAT- dependent YAP1 and Ctnnb1 activities advance our current understanding of mechanically stimulated $Ca^{2+}$ signaling in bone cells. The vertebrate bone is a $Ca^{2+}$ reservoir and highly regulated by both extra- and intracellular $Ca^{2+}$ homeostasis, which depends to a large extent on $Ca^{2+}$ channels. Several types of $Ca^{2+}$ channels exist. However, despite the established functions of $Ca^{2+}$ influx and calcineurin/NFAT signaling in promoting osteoblast differentiation and bone formation (*Winslow et al., 2006*; *Koga et al., 2005*; *Sun et al., 2005*; *Cao et al., 2017*), the physiological $Ca^{2+}$ entry mechanisms that evoke calcineurin/NFAT activation were largely unexplored in the past. The members of the TRPV and TRPP family of ion channels are involved in the extracellular calcium homeostasis and/or intracellular $Ca^{2+}$ signaling in response to mechanical stimuli in bone cells. However, physiological regulation of these ion channels in bone development and their roles in $Ca^{2+}$ signaling remain unclear, hampering mechanistic understanding of their functions at the cellular and molecular levels.

The Piezo1/2-dependent $Ca^{2+}$ signaling cascade in bone development we have identified opens the door to further understanding of many critical biological and medical processes in the skeletal system and beyond, given the importance of mechanotransduction and NFAT, Yap1, Ctnnb1 transcription factors in normal functions and diseases of many tissues and organs. For instance, how other $Ca^{2+}$/Calmodulin signaling targets such as CAMKII act in the PIEZO1/2 initiated mechanotransduction should be investigated. In addition, despite the crucial functions of NFATc1 in osteoclast differentiation, the mechanisms that modulate $Ca^{2+}$ oscillations during early osteoclastogenesis are still largely unknown. Although joint formation has been found to require movement and Piezo2 linage cells were found in the developing joints (*Figure 1d*), we did not observe abnormalities in joint formation in the *Piezo1* CKO and *Piezo1/2* DKO mutants. However, abnormal loading leads to cartilage damage and osteoarthritis as well as pathological bone alterations that may result in fracture when accumulated (*Astur et al., 2016*; *Keaveny et al., 1999*). Along this line, severe bone loss in astronauts due to the microgravity environment in space is a major obstacle of long-term space mission. In our preliminary study, expression of *PIEZO1* and *PIEZO2* in human BMSCs obtained from 11 male subjects showed a significant negative correlation with age (*Figure 11*, *Figure 11—source data 1*), suggesting that reduction of PIEZO channels might also contribute to bone aging. Further dissection of the Piezo1/2 function in a cell-type and age specific manner is likely to provide new insights into the underlying cellular and molecular mechanisms of mechanical forces in development and health.

## Materials and methods

**Key resources table**

| Reagent type (species) or resource | Designation | Source or reference | Identifiers | Additional information |
| --- | --- | --- | --- | --- |
| Gene (*M. musculus*) | *Piezo1* | | Gene ID: 234839 | |
| Gene (*M. musculus*) | *Piezo2* | | Gene ID: 667742 | |

*Continued on next page*

*Continued*

| Reagent type (species) or resource | Designation | Source or reference | Identifiers | Additional information |
|---|---|---|---|---|
| Strain, strain background (C57BL/6J, Female and male) | C57BL/6J Wild type mice | Jackson laboratory | Cat# JAX:000664, RRID:IMSR_JAX:000664 | |
| Strain, strain background (C57BL/6J, Female and male) | *Piezo1-tdTomato* mice | Jackson laboratory | Cat# JAX:029214, RRID:IMSR_JAX:029214 | |
| Strain, strain background (C57BL/6J, Female and male) | *Piezo1$^{fl/fl}$* mice | Jackson laboratory | Cat# JAX:029213, RRID:MSR_JAX:029213 | |
| Strain, strain background (C57BL/6J, Female and male) | *Piezo2$^{fl/fl}$* mice | Jackson laboratory | Cat# JAX:027720, RRID:IMSR_JAX:027720 | |
| Strain, strain background (C57BL/6J, Female and male) | *Piezo2-GFP-IRES-Cre* mice | Jackson laboratory | Cat# JAX:027719, RRID:IMSR_JAX:027719 | |
| Strain, strain background (C57BL/6J, Female and male) | *Ai9* mice | Jackson laboratory | Cat# JAX:007909, RRID:IMSR_JAX:007909 | |
| Strain, strain background (C57BL/6J, Female and male) | *Prrx1-Cre* mice | Jackson laboratory | Cat# JAX:005584, RRID:IMSR_JAX:005584 | |
| Strain, strain background (C57BL/6J, Female and male) | *Sp7-GFP::Cre* mice | Jackson laboratory | Cat# JAX:006361, RRID:IMSR_JAX:006361 | |
| Strain, strain background (C57BL/6J, Female and male) | *PC::GCaMP5G-tdTomato* mice | Jackson laboratory | Cat# JAX:024477, RRID:IMSR_JAX:024477 | |
| Cell line (*Homo sapiens*) | HEK 293T | ATCC | Cat# ACS-4500, RRID:CVCL_4V93 | |
| Transfected construct (*M. musculus*) | mPiezo1 siRNA | Sigma-Aldrich | Cat# SASI_Hs01_00208584 | |
| Transfected construct (*M. musculus*) | siRNA Universal Negative Control | Sigma-Aldrich | Cat# SIC001 | |
| Biological sample (*M. musculus*) | Primary BMSCs | | | Freshly isolated from *M. musculus* |
| Antibody | Anti-RFP (Rabbit polyclonal) | Rockland | Cat# 600-401-379, RRID:AB_2209751 | 1:200 (IHC), 1:1000 (WB) |
| Antibody | Anti-Sp7 (Rabbit polyclonal) | Abcam | Cat# ab22552, RRID:AB_2194492 | 1:600 (IHC), 1:2000 (WB) |
| Antibody | Anti-SO9 (Rabbit polyclonal) | Millipore | Cat# AB5535, RRID:AB_2239761 | 1:500 (IHC) |
| Antibody | Anti-OPN (Goat polyclonal) | R and D systems | Cat# AF808, RRID:AB_2194992 | 1:200 (IHC) |
| Antibody | Anti-Ctsk (Rabbit polyclonal) | Abclonal | Cat# A1782, RRID:AB_2763824 | 1:100 (IHC) |

*Continued on next page*

*Continued*

| Reagent type (species) or resource | Designation | Source or reference | Identifiers | Additional information |
|---|---|---|---|---|
| Antibody | Anti-Trap (Rabbit polyclonal) | Abclonal | Cat# A2528, RRID:AB_2764419 | 1:100 (IHC) |
| Antibody | Anti-BrdU (mouse monoclonal) | Biolengd | Cat# 317902, RRID:AB_604040 | 1:100 (IHC) |
| Antibody | Anti-Col10a1 (Rabbit polyclonal) | Abclonal | Cat# A6889, RRID:AB_2767448 | 1:100 (IHC) |
| Antibody | Anti-YAP/TAZ (Rabbit monoclonal) | CST | Cat# 8418, RRID:AB_10950494 | 1:300 (IF), 1:200 (IHC), 1:1000 (WB) |
| Antibody | Anti-β-catenin (mouse monoclonal) | BD Biosciences | Cat# 610154, RRID:AB_397555 | 1:400 (IHC), 1:2000 (WB) |
| Antibody | Anti-β-catenin (Rabbit polyclonal) | Abclonal | Cat# A0316, RRID:AB_2757122 | 1:1000 (WB) |
| Antibody | Anti-NFAT2 (Rabbit polyclonal) | Abclonal | Cat# A1539, RRID:AB_2762296 | 1:200 (IF), 1:100 (IHC), 1:1000 (WB) |
| Antibody | Anti-Phospho-YAP (Ser127) (Rabbit polyclonal) | CST | Cat# 4911, RRID:AB_2218913 | 1:1000 (WB) |
| Antibody | Anti-Phospho-β-catenin (Thr41/Ser45) (Rabbit polyclonal) | CST | Cat# 9565, RRID:AB_331731 | 1:1000 (WB) |
| Antibody | Non-phospho (Active) β-Catenin (Ser33/37/Thr41) (D13A1) Rabbit mAb | CST | Cat# 8814S, RRID:AB_11127203 | 1:1000 (WB) |
| Antibody | β-Actin (8H10D10) Mouse mAb | CST | Cat# 3700, RRID:AB_2242334 | 1:50,000 (WB) |
| Antibody | Anti-LATS1 (Goat polyclonal) | Santa Cruz | Cat# sc-9388, RRID:AB_2133367 | 1:1000 (WB) |
| Antibody | Anti-Phospho-LATS1 (Thr1079) (Rabbit monoclonal) | CST | Cat# 8654, RRID:AB_10971635 | 1:1000 (WB) |
| Antibody | Anti-FAK (mouse monoclonal) | Santa Cruz | Cat# sc-271126, RRID:AB_10614323 | 1:500 (WB) |
| Antibody | Anti-Phospho-FAK (pY397) (mouse monoclonal) | BD Biosciences | Cat# 611806, RRID:AB_399286 | 1:1000 (WB) |
| Antibody | Anti-Mst1 (Rabbit monoclonal) | CST | Cat# 14946, RRID:AB_2798654 | 1:1000 (WB) |
| Antibody | Anti-Phospho-Mst1 (Thr183) (Rabbit monoclonal) | CST | Cat# 49332, RRID:AB_2799355 | 1:1000 (WB) |
| Antibody | Anti-RUNX2 (Rabbit polyclonal) | ABclonal | Cat# A2851, RRID:AB_2764676 | 1:1000 (WB) |
| Antibody | Anti-CTGF (Rabbit polyclonal) | ABclonal | Cat# A11067, RRID:AB_2758390 | 1:1000 (WB) |
| Antibody | Anti-Calcineurin (Rabbit polyclonal) | Abclonal | Cat# A1063, RRID:AB_2758155 | 1:1000 (WB), 1:200 (IP) |
| Antibody | Anti-Flag (M2) (mouse monoclonal) | Sigma | Cat# F1804, RRID:AB_262044 | 1:2000 (WB), 1:1000 (IP) |
| Antibody | Anti-HA (3F10) (Rat monoclonal) | Roche | Cat# 11867423001, RRID:AB_390918 | 1:1000 (WB) |

*Continued*

| Reagent type (species) or resource | Designation | Source or reference | Identifiers | Additional information |
|---|---|---|---|---|
| Antibody | Anti-c-Myc (9E10) (Mouse monoclonal) | Santa Cruz | Cat# sc-40, RRID:AB_627268 | 1:1000 (WB) |
| Antibody | Anti-GAPDH (Rabbit monoclonal) | CST | Cat# 5174, RRID:AB_10622025 | 1:3000 (WB) |
| Antibody | Anti-DIG-AP conjugate (Sheep) | Roche | Cat# 11093274910, RRID:AB_514497 | 1:500 (ISH) |
| Antibody | Alexa Fluor 488 donkey anti-mouse (polyclonal) | Life Technologies | Cat# A-21202, RRID:AB_141607 | 1:500 (IHC) |
| Antibody | Alexa Fluor 488 donkey anti-rabbit (polyclonal) | Life Technologies | Cat# A-21206, RRID:AB_2535792 | 1:500 (IHC) |
| Antibody | Alexa Fluor 488 donkey anti-goat (polyclonal) | Life Technologies | Cat# A-11055, RRID:AB_2534102 | 1:500 (IHC) |
| Antibody | Alexa Fluor 568 donkey anti-mouse (polyclonal) | Life Technologies | Cat# A10037, RRID:AB_2534013 | 1:500 (IHC) |
| Antibody | Alexa Fluor 568 donkey anti-rabbit (polyclonal) | Life Technologies | Cat# A10042, RRID:AB_2534017 | 1:500 (IHC) |
| Antibody | Alexa Fluor 568 donkey anti-goat (polyclonal) | Life Technologies | Cat# A-11057, RRID:AB_142581 | 1:500 (IHC) |
| Antibody | ECL Donkey anti-rabbit | GE Healthcare Life Science | Cat# NA9340-1ml, RRID:AB_772191 | 1:5000 (WB) |
| Antibody | ECL Sheep anti-mouse | GE Healthcare Life Science | Cat# NA9310-1ml, RRID:AB_772193 | 1:5000 (WB) |
| Antibody | Bovine anti-goat IgG HRP | Santa Cruz | Cat# sc-2350, RRID:AB_634811 | 1:5000 (WB) |
| Antibody | Donkey Anti-Rat IgG Antibody | Sigma | Cat# AP189P, RRID:AB_11214462 | 1:5000 (WB) |
| Recombinant DNA reagent | pcDNA3.0-YAP-HA (plasmid) | This paper | | |
| Recombinant DNA reagent | pcDNA3.0-NFATc1-Flag (plasmid) | This paper | | |
| Recombinant DNA reagent | pcDNA3.0-β-catenin-Myc (plasmid) | This paper | | |
| Recombinant DNA reagent | pGL3-NFAT luciferase (plasmid) | Addgene | RRID:Addgene_17870 | |
| Recombinant DNA reagent | 8XGTIIC-luciferase (plasmid) | Addgene | RRID:Addgene_34615 | |
| Recombinant DNA reagent | Super 8X TOPFlash (plasmid) | Addgene | RRID:Addgene_12456 | |
| Recombinant DNA reagent | pTK-Renilla (plasmid) | Promega | Cat# E2241 | |
| Recombinant DNA reagent | pcDNA3.0-Calcineurin A-S (plasmid) | This paper | | |
| Sequence-based reagent | | | | Primers listed in supplemental table |

*Continued on next page*

Continued

| Reagent type (species) or resource | Designation | Source or reference | Identifiers | Additional information |
|---|---|---|---|---|
| Commercial assay or kit | Gel Extraction Kit | Omega | Cat# D2500 | |
| Commercial assay or kit | RNAscope 2.5 HD Reagent Kit | ACD | Cat# 322350 | |
| Commercial assay or kit | RNAscope Probe Mm-*Piezo1* | ACD | Cat# 500511 | |
| Commercial assay or kit | RNAscope Probe Mm-*Ctgf* | ACD | Cat# 314541 | |
| Commercial assay or kit | 1-Step NBT/BCIP Substrate Solution | Life Technologies | Cat# 34042 | |
| Commercial assay or kit | Ion 550 Chip Kit | Life Technologies | Cat# A34541 | |
| Commercial assay or kit | Ion AmpliSeq Transcriptome Mouse Gene Expression Panel, Chef-Ready Kit | Life Technologies | Cat# A36412 | |
| Commercial assay or kit | Dual-Luciferase Reporter Assay | Promega | Cat# E1910 | |
| Commercial assay or kit | Lipofectamine 3000 | Life Technologies | Cat# L3000001 | |
| Commercial assay or kit | RAT/Mouse P1NP ELISA kit | Immunodiagnostic Systems Inc | Cat# AC33F1 | |
| Commercial assay or kit | Alexa Fluor 594 Tyramide Super BoostTM Kit | Life Technologies | Cat# B40944 | |
| Commercial assay or kit | Masson's Trichrome | Abcam | Cat# ab150686 | |
| Commercial assay or kit | TRAP staining kit | Sigma-Aldrich | Cat# 387A | |
| Commercial assay or kit | TUNEL kit | Life Technologies | Cat# C10617 | |
| Commercial assay or kit | FAST SYBR Green Master Mix | Life Technologies | Cat# 4385612 | |
| Commercial assay or kit | High-Capacity cDNA Reverse Transcription Kit | Life Technologies | Cat# 4374966 | |
| Chemical compound, drug | DIG RNA labeling mix | Roche | Cat# 11277073910 | |
| Chemical compound, drug | Protector RNase inhibitor | Roche | Cat# 03335399001 | |
| Chemical compound, drug | T7 RNA polymerase | Roche | Cat# 10881767001 | |
| Chemical compound, drug | T3 RNA polymerase | Roche | Cat# 11031163001 | |
| Chemical compound, drug | LiCl | Sigma-Aldrich | Cat# L4408 | |
| Chemical compound, drug | Phusion High-Fidelity DNA Polymerase | New England Biolabs | Cat# M0530 | |

*Continued*

| Reagent type (species) or resource | Designation | Source or reference | Identifiers | Additional information |
|---|---|---|---|---|
| Chemical compound, drug | Proteinase K | Sigma-Aldrich | Cat# P2308 | |
| Chemical compound, drug | Glutaraldehyde solution | Sigma-Aldrich | Cat# G5882 | |
| Chemical compound, drug | Yeast tRNA | Sigma-Aldrich | Cat# R8759 | |
| Chemical compound, drug | Blocking reagent | Roche | Cat# 11096176001 | |
| Chemical compound, drug | BM-purple | Roche | Cat# 11442074001 | |
| Chemical compound, drug | Alizarin Red S | Sigma-Aldrich | Cat# A5533 | |
| Chemical compound, drug | Alcian Blue | Sigma-Aldrich | Cat# A9186 | |
| Chemical compound, drug | RNAzol RT | Sigma-Aldrich | Cat# R4533 | |
| Chemical compound, drug | Calcein | Sigma-Aldrich | Cat# C0875 | |
| Chemical compound, drug | Sliver nitrate | Sigma-Aldrich | Cat# S8157 | |
| Chemical compound, drug | Sodium thiosulfate | Sigma-Aldrich | Cat# 72049 | |
| Chemical compound, drug | BGjb medium | Life Technologies | Cat# 12591038 | |
| Chemical compound, drug | Yoda1 | TOCRIS | Cat# 5586 | |
| Chemical compound, drug | Cyclosporin A | LC Laboratories | Cat# LC-C-6000 | |
| Chemical compound, drug | Gadolinium (III) chloride | Sigma-Aldrich | Cat# 439770 | |
| Chemical compound, drug | β-glycerophosphate | Sigma-Aldrich | Cat# G9422 | |
| Chemical compound, drug | L-ascorbic acid | Sigma-Aldrich | Cat# A5960 | |
| Chemical compound, drug | Sulfo-SANPAH | Life Technologies | Cat# 22589 | |

*Continued on next page*

*Continued*

| Reagent type (species) or resource | Designation | Source or reference | Identifiers | Additional information |
|---|---|---|---|---|
| Chemical compound, drug | Collagen I, Rat tail | Corning | Cat# 354236 | |
| Chemical compound, drug | BIO | Sigma-Aldrich | Cat# B1686 | |
| Chemical compound, drug | 40% (w/v) acrylamide stock solution | Sigma-Aldrich | Cat# A4058 | |
| Chemical compound, drug | 2% (w/v) bis-acrylamide stock solution | Sigma-Aldrich | Cat# M1533 | |
| Software, algorithm | GraphPad Prism | GraphPad Prism | RRID:SCR_002798 | |
| Software, algorithm | ImageJ | ImageJ | RRID:SCR_003070 | |
| Software, algorithm | DAVID | DAVID | RRID:SCR_001881 | |
| Software, algorithm | Photoshop | Adobe | RRID:SCR_014199 | |
| Software, algorithm | Transcriptome Analysis Console | Life Technologies | RRID:SCR_016519 | |

## Mouse lines

All animal experiments were carried out according to protocols approved by the Harvard Medical School Institutional Animal Care and Use Committee. Mice described in the literature and purchased from the Jackson Laboratories: *Piezo1*$^{P1-tdT}$ (stock# 029214) (*Ranade et al., 2014*), *Piezo2-EGFP-IRES-Cre* (*Piezo2*$^{tm1.1(cre)Apat}$, stock# 027719) (*Woo et al., 2014*), *Piezo1*$^{f/f}$ (stock# 029213) (*Cahalan et al., 2015*), *Piezo2*$^{f/f}$ (stock# 027720) (*Woo et al., 2014*), *Prrx1-Cre* (stock# 005584)

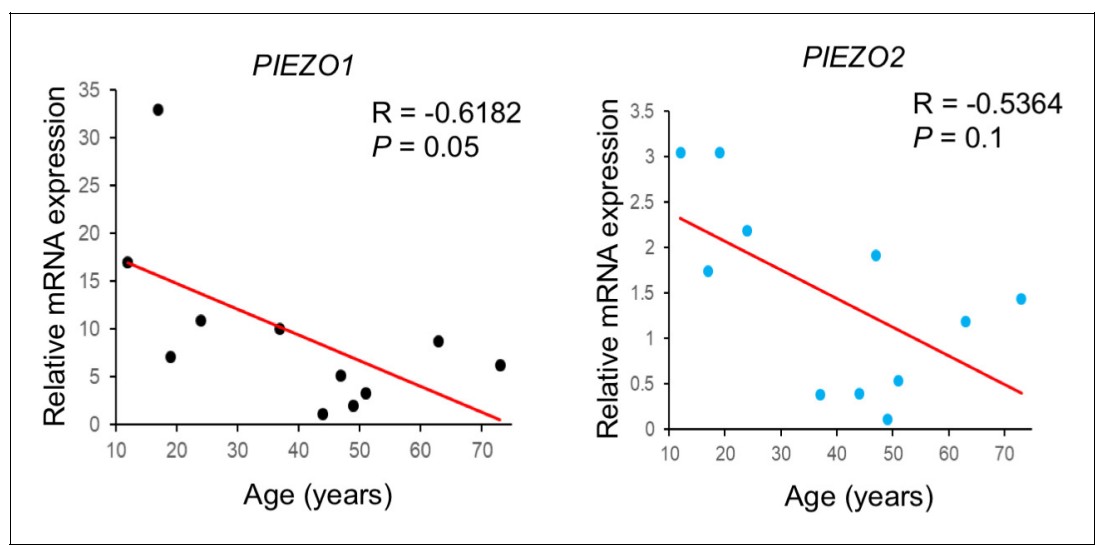

**Figure 11.** Expression of *POEZO1/2* in human BMSCs. QRT-PCR results using the cDNA of human BMSCs from 11 male subjects, between 12 to 75 years old. *PIEZO1/2* Expression levels were normalized to GAPDH expression. The correlation between the expression of *PIEZO1* or *PIEZO2* and age was then calculated by Spearman's correlation in Microsoft Excel (*Figure 11—source data 1*).

The online version of this article includes the following source data for figure 11:

**Source data 1.** Original numbers collected for quantification.

(*Logan et al., 2002*), *Sp7-GFP::Cre* (stock# 006361) (*Rodda and McMahon, 2006*), *Polr2a*$^{Tn(pb\text{-}CAG\text{-}GCaMP5g,\text{-}tdTomato)Tvrd}$ (*PC::G5-tdT,* stock# 024477) (*Gee et al., 2014*), *Ai9* (*Rosa26-TdTomato,* stock# 007909). For embryos or neonatal mice, both male and female were used in the analyses as sex could not be clearly identified in embryos or neonatal mice. Sex-matched littermate mice were compared in postnatal mice at day 21. Three or more littermate groups were examined and representative images are shown.

## RNA in situ hybridization

Whole mount RNA in situ hybridization was performed using digoxygenin-labeled anti-sense RNA probes as described before (*Wilkinson and Nieto, 1993*). The probes sequences were as described previously (mPiezo1 cDNA: base pairs 292–1054 of NM_001037298.1; mPiezo2 cDNA: base pairs 8,229–9298 of NM_001039485) (*Wu et al., 2016*). Tissue sections were subjected to in situ hybrization with the RNA Scope technology according to the manufacturer's instruction (Advanced Cell Diagnostics).

## Immunohistochemistry staining

Embryos and early-postnatal specimens were fixed in 4% (wt/vol) paraformaldehyde in phosphate-buffered saline (PBS) and processed for cryostat sections, which were blocked in 10% donkey serum and 0.1% Triton X-100 in PBS and Immunohistochemistry was performed with primary antibodies and secondary antibodies. Cryostat sections were mounted in mounting medium with DAPI from Vector laboratories (H-1200).

## Quantification of osteoblast differentiation by alkaline phosphatase staining

Osteoblast differentiation was assayed by alkaline phosphatase (ALP) staining and quantified with ImageJ software as previously described (*Dupont et al., 2011*). For each sample, at least five low magnification (X20) pictures were taken for each sample, and osteogenic differentiation was quantified by the ALP$^+$ area determined with ImageJ as the number of blue pixels across the picture. This value was normalized to the number of cells (Hoechst/nuclei) for each picture (arbitrary units).

## Skeletal preparation and μCT scanning

Alcian blue staining for cartilage and Alizarin red staining for mineralized tissues were performed as described (*Yang et al., 2003*). μCT scanning of long bones was performed using a SCANCO μCT 35 according to standard procedures.

## Bone histomorphometric analysis

Four weeks old wild type and *Prrx1-Cre*-driven *Piezo1* CKO or *Piezo1/2* DKO mice were subcutaneously injected with 20 mg/kg Calcein (Sigma) on days 4 and 1 before euthanization, respectively. Histomorphometric measurements were carried out semiautomatically with the OsteoMeasure image analyzer (OsteoMetric). Dynamic parameters including the mineralizing surface per bone surface (MS/BS) (percentage) and the bone formation rate per bone volume (BFR/BV) (percentage per year) were quantified. Analyses of bones from 3 mice per group were performed.

## Von Kossa staining and Masson's Trichrome staining

For von Kossa staining, cryostat sections were stained with 1% silver nitrate solution under a 60 W lamp for 1 hr. Slides were rinsed three times in distilled water. Sodium thiosulfate 5% was added to the slides for 5 min. Slides were rinsed three times in distilled water and counterstained with 0.1% nuclear fast red. Slides were rinsed three times in distilled water before mounted in mounting medium. For Masson's Trichrome staining, the sections were re-fixed in Bouin's solution for 1 hr at 56°C and then stained sequentially with Weigert's iron hematoxylin solution, Biebrish scarlet-acid fuchsin solution, and aniline blue solution. Collagen is stained blue.

## TUNEL assay

TUNEL assay was performed using a kit (Click-iT Plus TUNEL Assay for In Situ Apoptosis Detection, Life Technologies) according to the manufacturer's protocol.

## PINP ELISA analysis

Serum PINP levels were measured using a commercial kit from Immunodiagnostic Systems, according to the manufacturer's instructions.

## Cell culture

Mouse bone marrow stromal cells (BMSCs) were isolated as previously described with minor modifications (*Khan et al., 2018*). Briefly, femur and tibia bones were cut and the bone marrow was extracted under sterile conditions, dissociated and suspended in PBS and centrifuged immediately for 5 min at 300 g. The resulting pellet was suspended in culture medium (α-MEM supplemented with penicillin-streptomycin solution, sodium bicarbonate and 10% fetal bovine serum). BMSCs were cultured at 37°C and 5% $CO_2$. The third passage cells were used for experiments. BMSCs were seeded and cultured in 12-well plates with osteogenic differentiation medium to induce osteoblast differentiation as described previously (*Regard et al., 2013*). HEK293T cells were maintained under standard conditions (DMEM supplemented with 10% FBS and 100 U/mL penicillin/streptomycin).

## Mouse embryonic limb bud culture for bone and joint development

Mouse limbs were dissected from E13.5 embryos in PBS. The limbs were cultured as previously described (*Storm and Kingsley, 1999*). Briefly, the dissected fetal limbs were placed on a metal grid at the air-fluid interface with BGJb medium supplemented with 50 U/ml penicillin/streptomycin. Yoda1 (400 nM) was supplemented into the medium culturing right limbs, and equal volume of vehicle was supplemented into the medium culturing the left limbs (control). The limbs were maintained at 37°C in a 5% $CO_2$ incubator. The medium was changed every 2 days. The cultured limbs were harvested and subjected to analyses 4 days after being cultured.

## Transfection

After the cells attached to the culture dish and formed a monolayer with 70–80% confluency, the Ad-Cre or Ad-GFP virus (1:2000) was added to the medium. After 24 hr, the medium was changed, and the cells were cultured for another 24 hr before next step experiments. siRNA transfection was performed with Lipofectamine RNAi-MAX (Life Technologies) according to the manufacturer's instructions. siRNA (Piezo1: SASI_Hs01_00208584) and MISSION siRNA Universal Negative Control were purchased from Sigma. The following luciferase reporters, Ctnnb1 reporter Super 8X TOPFlash (#12456), Yap1 reporter 8XGTIIC (#34615) and NFATc1 reporter pGL3-NFAT (#17870), were purchased from Addgene.

## Mechanical loading

Oscillating fluid flow was generated using a horizontal bidirectional shaker (260350, Boekel Scientific Rocker II). BMSCs were cultured in osteogenic medium for 2 days followed by shaking (25 RPM; 20°; 12 hr shaking and 12 hr static) for 3 days (*Wittkowske et al., 2016*; *Choi et al., 2019*). Cell culture on hydrogels of different stiffness was performed as described (*Meng et al., 2018*; *Tse and Engler, 2010*).

## Unloading by BOTOX-induced paralysis

12-week-old wild type and *Sp7-Cre* driven Piezo1/2 DKO male mice were anesthetized by isoflurane and then injected with 0.5U botulinum toxin (BTX, List Biological Laboratories) into both the right quadriceps and the right calf muscles. This treatment is a tibial unloading model by muscle paralysis and disuse. 24 hr after BTX injection, mice were unable to use their right hind limb. Left tibiae served as normal loading controls. Since a rapid and profound bone loss could be observed 1 week after BTX injection (*Grimston et al., 2007*), the mice were euthanized 10 days after injection and the tibiae were analyzed by μCT. A 0.8 mm thick section spanning the proximal tibia metaphysis was analyzed as shown in previous studies (*Judex et al., 2004*).

## Quantitative real-time PCR

Total RNA from mouse humerus and femur tissues bone tissue devoid of bone marrow was prepared using the TRIZOL reagent (Life Technologies) or RNeasy Mini Kit (Qiagen) according to the manufacturer's protocols. cDNA was synthesized from total RNA (1–3 μg) using SuperScript II Reverse

Transcriptase with random primer (Life Technologies). QRT-PCR were performed using SYBR Select Master Mix on StepOnePlus thermal cycler from Applied Biosystems. Expression levels were always given relative to glyceraldehyde 3-phosphate dehydrogenase (Gapdh). The primer sequences of the genes are provided in the *Supplementary file 2*.

### RNA sequencing

Total RNA was isolated from humerus and femur tissue tissues at P0 pups as described above. The libraries were constructed using Ion AmpliSeq Transcriptome Mouse Gene Expression Panel, Chef-Ready Kit according to the manufacturer's protocols. The library qualities were checked by running on a BioAnalyzer 2100 and the concentrations were determined from the analysis profiles. Six barcoded libraries were pooled together on an equimolar basis and run using the Ion 550 Chip Kit. Genes with a change fold $\geq 2.82$ or $\leq -0.35$ were identified as differentially expressed genes and analyzed using the DAVID Bioinformatics Resources 6.8 (*Huang et al., 2007*).

### Calcium imaging

Primary BMSCs from the *PC::G5-tdT* or *Piezo1*$^{f/f}$; *PC::G5-tdT* mice were cultured as described above. GFP signal of GCaMP5G was excited at 480/505 nm and fluorescence emission was collected at 525 nm. Images were continuously taken by a Leica DM IRB microscope at time intervals of 5 s. Cells were stimulated either with shear force or Yoda1 (40 μM). All experiments were performed at room temperature (25°C). Fluorescence time series were analyzed using Image J software and converted to $\Delta F/F_0$ ($\Delta F/F_0 = (F - F_0)/F_0$), where $F_0$ is the baseline fluorescence intensity.

### Luciferase reporter assay

To measure the activity of YAP1/WWTR1, Wnt/Ctnnb1, and NFAT signaling, cells seeded in 96-well plates were cotransfected with the luciferase reporters of 8XGTIIC, 8XSuper Top-Flash or pGL3-NFAT, respectively, with pTK-*Renilla* (Promega) and effector plasmids. Luciferase activity was measured with a dual-luciferase reporter assay kit (Promega) according to the manufacturer's instructions.

### co-immunoprecipitation and immunoblotting

Bone tissues or cells were prepared using a lysis buffer [20 mM Tris (pH 7.4), 150 mM NaCl, 1% Triton X-100, 1 mM EDTA, 1 mM EGTA, 2.5 mM sodium pyrophosphate, 1 mM β-glycerophosphate, 1 mM sodium orthovanadate] or RIPA buffer (Santa Cruz Biotechnology), respectively, containing protease inhibitor mixture (Roche). Immnoprecipitates or total cell lysates were analyzed by Western blotting according to standard procedures.

### Small molecule treatment

BIO was prepared as described previously (*Sato et al., 2004*). Pregnant females were injected with Bio by intraperitoneal injection at a concentration of 2 μM every day from E15.5 and the postnatal mice were injected every other day. Equivalent volumes of vehicle were injected as control. Yoda1 (100 nM), $Gd^{3+}$ (100 nM), CsA (100 ng/mL) or Bio (1 μM) were added in cell culture medium after two days of osteogenic induction.

### Statistical analysis

All data analysis in this study was carried out using GraphPad Prism 7 (GraphPad Software). Quantifications were done from at least three independent experimental groups. Statistical analysis between groups was performed by two-tailed Student's *t* test to determine significance when only two groups were compared. One-way ANOVA with Tukey's post-hoc tests were used to compare differences between multiple groups. p-Values of less than 0.05 and 0.01 were considered significant. Error bars on all graphs are presented as the SD of the mean unless otherwise indicated.

### Study approval

The procedures for all animal experiments were reviewed and approved by the Harvard Medical School Institutional Animal Care and Use Committee.

## Acknowledgements

We are grateful to the Yang lab for stimulating discussions. YY, YF, YL, QC and PY are supported by the NIH grants R01DE025866 and R01AR070877. Taifeng Zhou, Xiaolei Zhang and Jiachen Lin are supported by visiting graduate student fellowships from the China Scholarship Council of the Chinese Ministry of Education.

## Additional information

### Funding

| Funder | Grant reference number | Author |
|---|---|---|
| National Institute of Dental and Craniofacial Research | R01DE025866 | Qian Cong Yingzi Yang |
| National Institute of Arthritis and Musculoskeletal and Skin Diseases | R01AR070877 | Prem S Yadav Yingzi Yang |
| National Cancer Institute | R01CA222571 | Yuchen Liu |
| China Scholarship Council | 201806380049 | Taifeng Zhou |
| China Scholarship Council | 201806210436 | Jiachen Lin |
| National Institute of Arthritis and Musculoskeletal and Skin Diseases | | Yi Fan |

The funders had no role in study design, data collection and interpretation, or the decision to submit the work for publication.

### Author contributions

Taifeng Zhou, Data curation, Formal analysis, Validation, Investigation, Methodology; Bo Gao, Yi Fan, Yuchen Liu, Qian Cong, Data curation, Formal analysis, Investigation, Methodology; Shuhao Feng, Data curation, Formal analysis, Investigation; Xiaolei Zhang, Prem S Yadav, Data curation, Investigation; Yaxing Zhou, Formal analysis, Methodology; Jiachen Lin, Investigation; Nan Wu, Liang Zhao, Dongsheng Huang, Resources, Supervision; Shuanhu Zhou, Resources, Investigation; Peiqiang Su, Resources, Supervision, Funding acquisition, Investigation, Project administration; Yingzi Yang, Conceptualization, Resources, Supervision, Funding acquisition, Validation, Investigation, Methodology, Project administration

### Author ORCIDs

Nan Wu http://orcid.org/0000-0002-9429-2889
Yingzi Yang https://orcid.org/0000-0003-3933-887X

### Ethics

Animal experimentation: This study was performed in strict accordance with the recommendations in the Guide for the Care and Use of Laboratory Animals of the NIH. All of the animals were handled according to approved institutional animal care and use committee (IACUC) protocols (#IS00000121-3) of the Harvard Medical School. The protocol was approved by the Committee on the Ethics of Animal Experiments of the Harvard Medical School.

### Decision letter and Author response

Decision letter https://doi.org/10.7554/eLife.52779.sa1
Author response https://doi.org/10.7554/eLife.52779.sa2

## Additional files

### Supplementary files

- Source data 1. Original numbers collected for quantification.
- Supplementary file 1. Quantified results of µCT scanning of the tibia bones from the wild-types control and *Sp7-Cre*-driven *Piezo1* and *Piezo1/2* mutant mice (*Source data 1*).
- Supplementary file 2. The sequences of oligo primers used in RT-PCR.
- Transparent reporting form

### Data availability

RNAseq source data for Figure 4 has been deposited in GEO under the accession number GSE139121. All data generated or analysed during this study are included in the manuscript and supporting files.

The following dataset was generated:

| Author(s) | Year | Dataset title | Dataset URL | Database and Identifier |
|---|---|---|---|---|
| Taifeng Zhou, Yu-chen Liu, Yingzi Yang | 2019 | RNA seq of femur and humerus bone tissues from Prx1cre driven Piezo1/2 mutant pups at the age of P0 | https://www.ncbi.nlm.nih.gov/geo/query/acc.cgi?acc=GSE139121 | NCBI Gene Expression Omnibus, GSE139121 |

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
