## [Decision Letter]

**Acceptance summary:**

This paper describes a new pathway for mechanotransduction in bone and its relevance during skeletal development. This involves the activation of Piezo 1/2 channels in osteoblast progenitors, which leads to calcium signals that activate calcineurin with further downstream effects on YAP1, NFATc1 and β-catenin to regulate bone formation and resorption.

**Decision letter after peer review:**

Thank you for submitting your article "Piezo1/2 mediate mechanotransduction essential for bone formation through concerted activation of NFAT-YAP-β-catenin" for consideration by *eLife*. Your article has been reviewed by two peer reviewers, and the evaluation has been overseen by Mone Zaidi as the Reviewing Editor and Clifford Rosen as the Senior Editor. The reviewers have opted to remain anonymous.

The reviewers have discussed the reviews with one another and the Reviewing Editor has drafted this decision to help you prepare a revised submission.

Summary:

The study explores a role for the Piezo proteins in bone formation using *Prrx1* and *Osx* as the Cre tools. Piezo1 and Piezo2 were found to be expressed in long bones during embryonic development and to regulate osteoblastogenesis in response to mechanical forces. The experiments are methodologically sound, carefully conducted and well controlled. They provide major new insights, and are explained in a well crafted and logically presented manuscript. With that said, the Reviewers and Editors have expressed concerns that are summarized below. These concerns would need to be resolved in a revised version.

Essential revisions:

1) There is a concern about the overlap of the present manuscript and two prior publications in *eLife* from other groups, notably Sun et al., 2019 and Li et al., 2019), which document a role for mechanosensitive-responsive Piezo1/2 channels in bone formation. While it was recognized that those past reports were not as technically sophisticated as the present submission, there is a concern regarding novelty. That said, the reviewers recognized that the present study, apart from being cutting-edge, provided new insights into areas of skeletal development. It is therefore critical that the authors highlight how specifically this study adds new dimensions to previous work and moves the field forward.

2) There is the missed opportunity of making the manuscript more distinctive. For example, there is a clear difference in the sites of bone fracture, notably limb versus rib, in in Piezo1 CKOs driven by *Prrx1* and *Osx Cre*, respectively. Studies examining the expression pattern of the Piezo1/2 proteins in sorted *Prrx1*-positive and *Osx*-positive osteoblast progenitors during differentiation under induced mechanotransduction conditions could shed light on this difference.

3) It was felt that the connection between Yap and Piezo activity was very interesting and was likely to be important for the growing Hippo-Yap field. However, insufficient evidence was presented to establish that activated Yap pathways contribute to osteoblast differentiation through Piezo. It was also unclear how precisely non-canonical Yap signaling plays a role – specifically, molecular mechanisms remained scanty. The authors would need to answer several key questions: (a) what are the respective functions of Yap1 and β-catenin in osteogenesis mediated by Piezo1/2; (b) what is the physiological function the NFAT/YAP/β-catenin complex; (c) what is the contribution of YAP1 to the rescue in Piezo1 and Piezo1/2 mutants by enhanced β-catenin activity; and (d) whether there is crosstalk between Wnt/β-catenin and Hippo/Yap signaling? The authors should comment in more depth on potential Hippo signaling in their system. This possibility is suggested by their data but does not come out well in the Discussion.

4) The authors show Piezo1/2 expression in long bone, but not in the axial skeleton or the skull. Specifically, are there abnormalities in craniofacial bones or teeth?

5) BTX treatment is not the best means to examine the mechanosensor function of Piezo proteins in bone formation, as injured muscle and nerves are known to alter bone formation indirectly. While the hind-limb suspension model or tibia axial loading would provide a better assessment, it would suffice to provide full histomorphometric analysis to confirm the function of the Piezo proteins in bone formation.

---

## [Author Response]

Essential revisions:1) There is a concern about the overlap of the present manuscript and two prior publications in eLife from other groups, notably Sun et al., 2019 and Li et al., 2019), which document a role for mechanosensitive-responsive Piezo1/2 channels in bone formation. While it was recognized that those past reports were not as technically sophisticated as the present submission, there is a concern regarding novelty. That said, the reviewers recognized that the present study, apart from being cutting-edge, provided new insights into areas of skeletal development. It is therefore critical that the authors highlight how specifically this study adds new dimensions to previous work and moves the field forward.

Thanks for the compliments that *“*the present study, apart from being cutting-edge, provided new insights into areas of skeletal development”. We have read previous reports carefully: Sun et al., 2019, Li et al., 2019 and Wang et al., 2020. We also appreciate the suggestion from the reviewers and editors. We have now highlighted how specifically in our study adds new dimensions to previous work and moves the field forward in the Introduction, and Discussion, which are also summarized as follows:

1) While all of the recently published studies focus on postnatal bone and regulation of mature osteoblast/osteocyte functions by the mechanical force channel Piezo1, our studies also tackle the roles of both Piezo1 and 2 in early skeletal (cartilage and bone) development. We have stated in the Introduction that “mechanical forces are part of environmental cues that are sensed and responded to during embryonic development and adult life for proper morphogenesis and tissue/organ functions” and “despite the long-recognized key regulatory roles of mechanotransduction for embryonic development and sensory perception, little concrete information on the molecular mechanisms that integrate biophysical stimuli with gene regulation to guide fundamental events such as cell fate determination, proliferation and apoptosis is known”. Our study therefore emphasized on osteoblast differentiation from embryonic gmesenchymal progenitor cells and adult bone marrow stromal cells (BMSCs) where stem cells are found. Our work uncovered the temporal and spatial expression of Piezo1 and 2 in embryonic long bones in situ and their roles in osteoblast differentiation from embryonic mesenchymal progenitor cells or BMSCs. In addition, despite the lower expression levels of *Piezo2* in the bone forming cells, it is important to know its functions. Our findings indicate that the role of Piezo2 in the skeleton should not be ignored. Despite the lack of apparent phenotypes in skeletal development in the *Piezo2* mutants, *Piezo2* loss did enhance the long bone phenotypes of *Piezo1* mutants. Our studies of Piezo1/2 in early embryonic development and BMSCs together advance the current mechanistic understanding of mechanic force sensation and response in cell fate decision from progenitor or stem cells. Importantly, our studies provide new opportunities to understand how mechanical information from environment in the form of biophysical stimuli such as stress, strain and fluid flow generated by gravity, cell movement and cell-cell or cell-extracellular matrix (ECM) interactions, join well-known complex networks of biochemical signaling that direct the development of higher vertebrate embryos as well stem cell regulation during tissue regeneration.

2) It is satisfying to see that reduction of Yap1 activities by loss of Piezo1 function was shown in two of the three recently published work. Our study specifically advanced current understanding of Yap1 activation in mechanotransduction by connecting Piezo1/2 mediated Ca^2+^ signaling with a signaling cascade leading to downstream actomyosin cytoskeleton rearrangement and a concerted activation of key transcription factors including Yap1, β-catenin (Ctnnb1) and NFAT, all of which have been demonstrated to play key roles in promoting osteoblast differentiation. Our finding that the Piezo channels regulate noncanonical Yap1 activation independently of the Mst (Stk3 and 4)/Lats kinase cascade open new doors to further understanding regulation of not only Yap1, but also β-catenin and NFAT in many other physiological and pathological contexts, given the importance of mechanotransduction and these transcription factors in the normal functions and diseases of many tissues and organs. More details are provided in our response to Point 4.

3) All 4 studies include ours provided a good opportunity to compare notes and clarify some issues in the future. For example, three studies (Sun et al., 2019, Li et al., 2019, and ours showed that loss of Piezo1 in osteoblast lineage cells reduced bone formation and osteoblast differentiation, while Wang et al., 2020 indicated that loss of Piezo1 in osteoblast cells increased bone resorption without reducing bone formation.

2) There is the missed opportunity of making the manuscript more distinctive. For example, there is a clear difference in the sites of bone fracture, notably limb versus rib, in in Piezo1 CKOs driven by Prrx1 and Osx Cre, respectively. Studies examining the expression pattern of the Piezo1/2 proteins in sorted Prrx1-positive and Osx-positive osteoblast progenitors during differentiation under induced mechanotransduction conditions could shed light on this difference.

Thanks for the comments. As we pointed out in the manuscript, the *Prrx1Cre* and *OsxCre* (*Sp7-Cre*) are expressed in overlapping but different tissues and cells. In the limb, *Prrx1Cre* is expressed in all mesenchymal progenitor cells that give rise to cartilage, bone and tendon/ligament. Its activity is also found in the limb muscle, while *Sp7-Cre* expression is limited to osteoblast lineage cells and hypertrophic chondrocytes. However, *Sp7* is expressed in all developing bones, many of them do not express *Prrx1*, for instance, the rib, which explains lack of rib phenotypes in the *Prrx1Cre* driven mutants. In the developing limb, tendon, ligament and muscle play critical roles transmitting the forces between skeletal and muscular tissues, and this function might require extra-skeletal Piezo1 function. This may explain why the *Prrx1Cre* driven *Piezo1* mutants resulted in more severe bone defects. To further understand the difference between the *Prrx1*-positive and *Sp7*-positive osteoblast progenitors, we have also sorted the Prrx1-positive BMSCs and Sp7-positive osteoblast progenitors, cultured them and determined Piezo1/2 expression during osteoblast differentiation under induced mechanotransduction conditions. The protein levels were not high enough to be reliably detected in the cultured cells by Western blotting, but we found that mRNA expression of *Piezo1* was slightly upregulated by fluid stress in the Prrx1 positive, but not Sp7 positive cells. There was no significant difference in *Piezo2* expression, and its expression levels were much lower than those of *Piezo1*. The data have been added as the revised Figure 4D and p11 in the revised manuscript.

3) It was felt that the connection between Yap and Piezo activity was very interesting and was likely to be important for the growing Hippo-Yap field. However, insufficient evidence was presented to establish that activated Yap pathways contribute to osteoblast differentiation through Piezo. It was also unclear how precisely non-canonical Yap signaling plays a role – specifically, molecular mechanisms remained scanty. The authors would need to answer several key questions: (a) what are the respective functions of Yap1 and β-catenin in osteogenesis mediated by Piezo1/2; (b) what is the physiological function the NFAT/YAP/β-catenin complex; (c) what is the contribution of YAP1 to the rescue in Piezo1 and Piezo1/2 mutants by enhanced β-catenin activity; and (d) whether there is crosstalk between Wnt/β-catenin and Hippo/Yap signaling? The authors should comment in more depth on potential Hippo signaling in their system. This possibility is suggested by their data but does not come out well in the Discussion.

Thanks for agreeing that the connection between Yap and Piezo activity was very interesting and was likely to be important for the growing Hippo-Yap field. We have performed additional experiments and added discussion in our data interpretation to make our points more clear. Here are our answers to the raised questions:

a) What are the respective functions of Yap1 and β-catenin in osteogenesis mediated by Piezo1/2?

The phenotypes of neonatal bone fractures, reduced bone formation and increased bone resorption found in the *Prrx1-Cre; Piezo1*, *Prrx1-Cre; Piezo1/*2 or *Sp7-Cre; Piezo1*, *Sp7-Cre; Piezo1/2* mutants were similar to those found in the *Yap1/Taz* and *β*-*catenin* mutants. Our rescuing experiments with BIO treatment (revised Figure 10 and Figure 10—figure supplement 1) indicate that at least part of the functions of Piezo1/2 induced Yap1 activation is to promote β-catenin activation. However, the incomplete phenotypic and Yap1 expression rescue by BIO treatment suggest that Yap1 has non-overlapping functions with β-catenin. A possible scenario is that Yap1 and β-catenin each have their unique targets to promote bone formation aside from enhancing each other’s activities. In addition, each also has its own parallel upstream regulatory pathways such as Piezo1/2 as explained in detail in (c). We have also performed new experiments (revised Figure 10—figure supplement 1C) as explained in the following (c) and (d) to make these points more clear in subsection “Enhanced Ctnnb1 activities partially rescued the bone defects of *Piezo1/2* bone mutants” and Discussion paragraph three.

b) What is the physiological function the NFAT/YAP/β-catenin complex?

Our data indicate that the physiological function of the complex is to allow calcineurin-activated NFAT to coordinately activate YAP and β-catenin. In addition, such complex formation indicate that Yap was activated together with β-catenin, suggesting that β-catenin can also be activated by mechanotransduction via Yap1 and may mediate some of Yap1 activity in promoting bone formation. This point has been made more clear in Discussion paragraph three.

c) What is the contribution of YAP1 to the rescue in Piezo1 and Piezo1/2 mutants by enhanced b-catenin activity?

It is very likely that Yap1 and β-catenin each have their unique targets to promote bone formation aside from enhancing each other’s activities. In addition, each also has its own upstream regulatory pathways. While BIO treatment strongly upregulated β-catenin protein levels, the increase of Yap1 was less appreciable (Figure 10C), suggesting that while β-catenin activation is essential for osteoblast differentiation and can promote Yap1 activation (revised Figure 9—figure supplement 1A and revised Figure 10—figure supplement 1C), Yap1 is also regulated by Piezo1/2 via β-catenin-independent pathway(s). In addition, blocking Yap1 transcription activity by Verteporfin (VP)(Liu-Chittenden et al., 2012) reduced the effects of BIO in promoting both β-catenin and Yap levels, suggesting that β-catenin requires Yap1 activity to be fully upregulated, consistent with previous published results (Pan et al., 2018). It is possible that the incomplete Yap1 protein rescue by BIO treatment contributed to the incomplete phenotypic rescue of the *Piezo1* and *Piezo1/2* mutant by BIO treatment (revised Figure 10). It is the concerted activation of NFATc1, Yap and β-catenin that constitutes fully functional mechanotransduction pathways downstream of Piezo channels in promoting bone formation. More results and discussion have been added in subsection “Enhanced Ctnnb1 activities partially rescued the bone defects of *Piezo1/2* bone mutants”.

d) whether there is crosstalk between Wnt/β-catenin and Hippo/Yap signaling?

Yes, definitely! The cross talk between Wnt/β-catenin and Hippo/Yap1 signaling have been shown by a number of previous publications. The exact nature of such crosstalk is likely to be context dependent and several crosstalk mechanisms may exist in parallel. In our study, the crosstalk was shown in revised Figure 9—figure supplement 1A. While β-catenin promoted Yap transcription activity, Yap also promoted β-catenin transcription activity. In addition, we also further investigated the crosstalk between endogenous Wnt/β-catenin and Hippo/Yap1 signaling in primary BMSCs during the revision (revised Figure 10—figure supplement 1C and subsection “Enhanced Ctnnb1 activities partially rescued the bone defects of *Piezo1/2* bone mutants”). We showed that while Wnt/β-catenin signaling activation by BIO increased β-catenin protein levels and reduced β-catenin phosphorylation as expected, BIO treatment also increased Yap1 protein levels and reduced Yap1 phosphorylation. In addition, reduction of Yap1 phosphorylation by treating the cells with XMU-MP-1, an inhibitor of Hippo kinase Stk3/4 (or Mst_1/2_) (Fan et al., 2016), was also accompanied by increase in total and non-phosphorylated β-catenin protein levels, supporting a positive interaction between the Wnt/β-catenin and Hippo/Yap1 signaling pathways in the BMSCs. In addition, blocking Yap1 transcription activity by Verteporfin (VP)(Liu-Chittenden et al., 2012) reduced the effects of BIO in promoting both β-catenin and Yap1 levels, suggesting that β-catenin requires Yap1 activity to be fully upregulated (revised Figure 10—figure supplement 1C).

4) The authors show Piezo1/2 expression in long bone, but not in the axial skeleton or the skull. Specifically, are there abnormalities in craniofacial bones or teeth?

Like in the long bone, our preliminary results showed Piezo1 is also expressed in the osteoblast progenitor and differentiated osteoblast cells in the skull. *Piezo2* is not expressed in osteoblast lineage cells in the developing cranial bone. However, we did not see consistent bone formation difference between the *Piezo1* or *Piezo2* mutants and the wild type controls at developmental or early postnatal stages. These data are consistent with similar findings in Wang et al., 2020. We have not yet examined Piezo1/2 expression in the developing teeth or spine, nor the phenotypes in the absence of Piezo1/2. These will be interesting topics for future thorough investigations.

5) BTX treatment is not the best means to examine the mechanosensor function of Piezo proteins in bone formation, as injured muscle and nerves are known to alter bone formation indirectly. While the hind-limb suspension model or tibia axial loading would provide a better assessment, it would suffice to provide full histomorphometric analysis to confirm the function of the Piezo proteins in bone formation.

It is known that muscle contractions and external gravitational loading are two primary sources of mechanical forces exerted on bone. It is necessary to test whether Piezo1/2 in osteoblast cells are required to sense mechanical forces exerted by muscle contractions using the established BTX injection model, while we agree that the BTX model has its limitations. The role of *Piezo1* in an external gravitational loading (the hindlimb suspension (HS) model) and tibia axial loading have been shown by three recent papers published in *eLife* (Sun et al., 2019, Li et al., 2019, Wang et al., 2020). We are aware that there are indirect detrimental effects of BTX treatment on muscles, but at least some of these effects are not specific to the injected limbs and also found in the contralateral control limbs (Ellman et al., Calcif Tissue Int. 2014). By using the contralateral leg as the control, much of the indirect effects in the BTX model could be normalized. We would like to mention that none of the current loading and unloading models are perfect. Muscle atrophy is also found in the HS models and tibia axial loading is performed under artificial conditions. Studies in all these models collectively provide complementary insights into the roles of Piezo1/2.

While the BTX model is relevant in testing the role of Piezo1/2 in osteoblast mechanosensing (revised Figure 5C), we feel it is not feasible and necessary to confirm with full histomorphometric analysis the function of the Piezo proteins in bone formation within two months for the following reasons: 1, We have shown in multiple assays including histomorphometric analysis that bone formation was severely reduced in the *Prrx1-Cre-driven Piezo1/2* mutant mice in the manuscript (Figure 2E, F). As defects in bone formation in the HS and tibia axial loading models have been demonstrated by histomorphometric analysis in the less severe *Piezo1* osteoblast-specific mutants (Sun et al., 2019, Li et al., 2019), it is less likely that reduced bone formation did not contribute to blunted bone loss in the BTX-injected, more severe *Sp7-Cre* driven *Piezo1/2* mutants. 2, We already found that loss of *Piezo1/2* in early osteoblast cells led to very severe reduction of bone formation (Figure 2E, F), such that dynamic bone formation parameters was not quantifiable in our *Piezo* mutants. Lastly, I have talked to our colleague Dr. Roland Barron, an expert in histomorphometric analysis and the core at MGH and both told us that it is not possible for them to finish sample preparation and analyses within 2 months even if the mice were ready for BTX injection two months ago. We were told that the reasonable estimate is 4-6 months given the need for mouse breeding and their current work load.